# Structure of the bacterial plant-ferredoxin receptor FusA

Rhys Grinter[1,2,3], Inokentijs Josts[1], Khedidja Mosbahi[1], Aleksander W. Roszak[4], Richard J. Cogdell[4], Alexandre M.J.J. Bonvin[5], Joel J. Milner[6], Sharon M. Kelly[4], Olwyn Byron[6], Brian O. Smith[4] & Daniel Walker[1]

Iron is a limiting nutrient in bacterial infection putting it at the centre of an evolutionary arms race between host and pathogen. Gram-negative bacteria utilize TonB-dependent outer membrane receptors to obtain iron during infection. These receptors acquire iron either in concert with soluble iron-scavenging siderophores or through direct interaction and extraction from host proteins. Characterization of these receptors provides invaluable insight into pathogenesis. However, only a subset of virulence-related TonB-dependent receptors have been currently described. Here we report the discovery of FusA, a new class of TonB-dependent receptor, which is utilized by phytopathogenic *Pectobacterium* spp. to obtain iron from plant ferredoxin. Through the crystal structure of FusA we show that binding of ferredoxin occurs through specialized extracellular loops that form extensive interactions with ferredoxin. The function of FusA and the presence of homologues in clinically important pathogens suggests that small iron-containing proteins represent an iron source for bacterial pathogens.

[1] Institute of Infection, Immunity and Inflammation, College of Medical, Veterinary and Life Sciences, University of Glasgow, Glasgow G12 8QQ, UK. [2] Institute of Microbiology and Infection, School of Immunity and Infection, University of Birmingham, Birmingham B15 2TT, UK. [3] Infection and Immunity Program, Biomedicine Discovery Institute and Department of Microbiology, Monash University, Clayton, Victoria 3804, Australia. [4] Institute of Molecular, Cell and Systems Biology, College of Medical, Veterinary and Life Sciences, University of Glasgow, Glasgow G12 8QQ, UK. [5] Bijvoet Center for Biomolecular Research, Faculty of Science, Utrecht University, Utrecht 3584 CH, The Netherlands. [6] School of Life Sciences, College of Medical, Veterinary and Life Sciences, University of Glasgow, Glasgow G12 8QQ, UK. Correspondence and requests for materials should be addressed to R.G. (email: rhys.grinter@monash.edu) or to D.W. (email: daniel.walker@glasgow.ac.uk).

The central role of iron in the electron transfer reactions of cellular redox chemistry and its insolubility under oxygenic conditions makes it a generally limiting nutrient for microbial growth[1]. Eukaryotic organisms exploit this limited availability, via a mechanism termed 'nutritional immunity', hindering the growth of pathogenic microbes by tightly sequestering iron within specialized proteins[2,3]. To counter this, microbes have developed specialized systems for liberating and importing iron from host proteins[4]. In Gram-negative bacteria, outer membrane receptors of the TonB-dependent receptor (TBDR) family fulfil this role by binding microbial iron-scavenging siderophores and iron-containing host proteins such as lactoferrin, transferrin and haemoglobin[5]. TBDRs interact with their substrates through a highly specialized extracellular structure, formed by the outer loops of a 22-stranded transmembrane β-barrel. After these initial interactions, this barrel provides a conduit for the iron or iron-siderophore complex to cross the outer membrane[6]. As illustrated by structural and evolutionary analysis of the TonB-dependent transferrin receptor from the genus Neisseria, these systems play an important role in pathogenesis and represent part of the evolutionary arms race between host and pathogen[7–9]. In contrast to our understanding of protein-binding TBDRs from mammalian pathogens, there have been no reports of specialist receptors utilized by plant pathogens to obtain iron from host proteins during infection[10].

Previously, we reported the discovery and subsequent structural characterization of the pectocins, an unusual class of colicin-like bacteriocins. Colicin-like bacteriocins are protein toxins produced by Gram-negative bacteria mostly for intraspecies or intragenus competition and often parasitize TBDRs to gain entry to their target cell[11,12]. Pectocins M1 and M2, which are produced by phytopathogenic Pectobacterium spp. for intra-species competition, contain a cytotoxic domain that is active against the cell wall precursor lipid II in the periplasm, fused to an iron-containing plant-like ferredoxin that acts as a receptor-binding domain[13]. During our characterization of the pectocins, it became apparent that in addition to being susceptible to a ferredoxin-containing bacteriocin, Pectobacterium spp. are also able to utilize plant ferredoxins as an iron source under iron-limiting conditions[14]. Moreover, competition experiments showed that both the pectocins and ferredoxin are bound by the same receptor during cell entry[15].

In our current work we have identified the outer membrane receptor responsible for ferredoxin and pectocin import in Pectobacterium spp., which we designate FusA. To understand the mechanism of ferredoxin import we have solved the crystal structure of FusA and two of its plant ferredoxin substrates, and using nuclear magnetic resonance (NMR)-driven molecular docking we have modelled the FusA–ferredoxin complex. In addition, through bioinformatic analysis, we show that FusA homologues are widespread in members of Enterobacteriaceae that form commensal or pathogenic associations with mammalian hosts. This suggests that this family of TBDRs also plays role in iron acquisition from the mammalian host.

## Results

**Identification of the pectocin M1 receptor in Pectobacterium.** To identify the outer membrane receptor of the ferredoxin uptake system, which we designated FusA, we isolated proteins from the outer membrane of the pectocin M1-sensitive Pectobacterium strain Pectobacterium atrosepticum LMG2386 and applied them to a nickel affinity column pre-loaded with His6-tagged pectocin M1. After elution of bound protein we observed a protein on SDS–PAGE at ∼100 kDa, which co-purified with pectocin M1 (Supplementary Fig. 1A). Peptide mass fingerprinting identified a 97 kDa TBDR PCC21_007820 from Pectobacterium carotovorum (Pbc) subsp. carotovorum PCC21 as the closest match (Supplementary Fig. 1B). To confirm this protein interacts with the pectocins, we repeated this experiment with the outer membrane fraction from Escherichia coli recombinantly expressing FusA (PCC21_007820), showing that recombinant FusA also interacts with pectocin M1 and M2 (Supplementary Fig. 1C). We then constructed a ΔfusA mutant using P. atrosepticum LMG2386 and determined its sensitivity to pectocin M1. In contrast to the parent strain, the ΔfusA mutant shows complete resistance to pectocin M1, with complementation of fusA restoring sensitivity (Fig. 1a). Thus, FusA is the receptor for the ferredoxin domain containing bacteriocin pectocin M1. As we have previously shown that pectocin M1 and spinach ferredoxin compete for binding to the same receptor, we also propose that FusA is also a plant ferredoxin receptor[15].

Bioinformatic analysis of this newly identified receptor shows that closely related homologues of fusA are found in all other sequenced strains of P. atrosepticum and P. carotovorum (>75% amino-acid sequence identity) and strains of the related soft rot pathogens of Dickeya species (>60% identity). In all cases, fusA is found in a putative operon with three additional genes that encode a TonB-like protein, a predicted periplasmic M16 protease

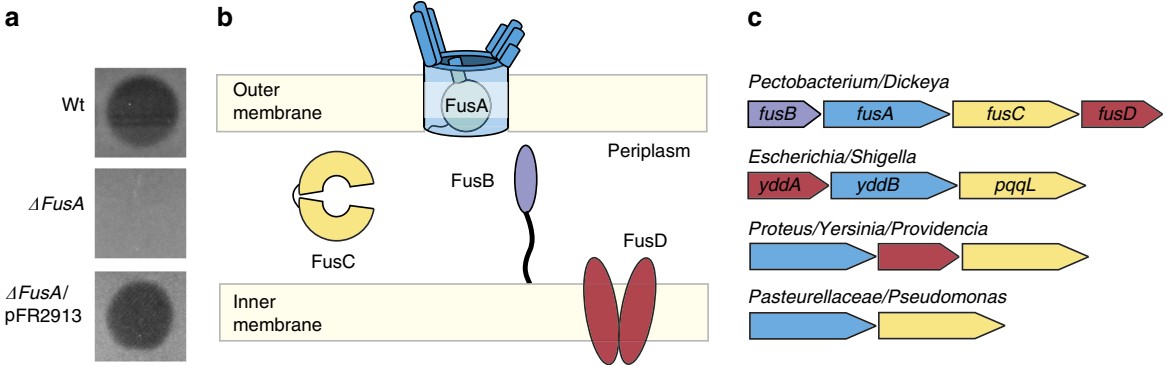

**Figure 1 | FusA from Pectobacterium is responsible for susceptibility to the ferredoxin domain containing bacteriocin pectocin M1** (**a**) Purified pectocin M1 spotted onto iron-limiting Lysogeny broth (LB) agar plus IPTG overlaid with a lawn of soft agar seeded with P. atrosepticum LMG2386, Wt, ΔfusA or ΔfusA complemented with a plasmid derived copy of fusA with an IPTG inducible promoter. Genetic knockout of fusA totally abolishes susceptibility of the strain to pectocin M1, while complementation of the knockout strain with plasmid encoded fusA restores the pectocin M1 susceptible phenotype (experiment repeated >6 times). (**b**) A schematic of the proteins of the Fus operon showing their predicted cellular localization and function. (**c**) The genetic organization of the putative Fus operon in Pectobacterium spp. and homologous genes from a subset of human pathogens.

and a fused ABC transporter that we refer to as *fusB*, *fusC* and *fusD*, respectively (Fig. 1b). In *Dickeya dadantii*, homologues of *fusA* and *fusC* are among the most highly upregulated genes during plant infection[16]. Interestingly, more distantly related homologues of the *fus* operon were also identified in bacteria that colonize and cause disease in mammalian hosts. These species include *E. coli* and members of the genera *Neisseria*, *Shigella*, *Yersinia*, the family Pasteurellaceae, as well as more distantly related members of β, δ and ε-proteobacteria (Fig. 1c, Supplementary Fig. 2 and Supplementary Table 1). Many of these species renowned for their utilization of specialized TBDRs and associated proteins to obtain iron from their hosts[6,17]. In *Pasteurella multocida*, *fusC* is upregulated in response to iron limitation and during infection[18,19], and in uropathogenic *E. coli*, homologues of FusA (YddB) and FusC (PqqL) are important for fitness in systemic infection[20]. The presence of genes encoding FusA homologues in bacteria associated with infection of diverse eukaryotic hosts suggests that the use of small iron-containing proteins as an iron source during infection extends beyond *Pectobacterium* spp.

**The crystal structure of the ferredoxin receptor FusA.** To determine the structural basis of ferredoxin binding by FusA we solved the structure of FusA by X-ray crystallography. Recombinant FusA (derived from *P. atrosepticum* SCRI1043) was expressed in *E. coli* and purified and refolded from inclusion bodies to yield stable, monodisperse FusA, which was used to undertake crystallization trials (Supplementary Fig. 3). Sparse matrix screening yielded crystals in a number of conditions and after extensive optimization crystals were obtained that diffracted beyond 3.2 Å at a synchrotron radiation source (Supplementary Fig. 4A,B). Phases were obtained and the structure solved using platinum single-wavelength anomalous dispersion data at 4.2 Å from a $K_2PtCl_4$ soaked crystal, with the final atomic model refined against native data to 3.2 Å resolution (Supplementary Figs 4C,D,5 and Table 1). FusA possesses a classical TBDR-fold, consisting of an N-terminal plug domain surrounded by a 22-stranded transmembrane β-barrel. However, the similarity of FusA to other TBDRs is limited to this transmembrane region, while the extracellular loops, which constitute ∼46% of the polypeptide chain, are highly divergent from any previously described protein structure (Fig. 2a). These loops largely consist of β-sheet and non-regular secondary structure elements, in addition to three small helical regions (Fig. 2b and Supplementary Fig. 6). Loops 4, 5 and 7 (from the N terminus of the barrel) are the most extended and form a wall-like β-sheet, which is separated from the barrel β-sheet by a section of irregular secondary structure. Large extensions are present at the end of loops 4 and 5, which double back over the outside of the barrel (Fig. 3). On the opposite side of FusA, loops 8–11 are also highly elongated and form extensive interactions with each other, although they lack the regular β-sheet structure of loops 4, 5 and 7 (Fig. 3). Normal mode analysis (NMA) indicates that the structures formed by these loops move between an open and closed conformation giving the extracellular portion of FusA the appearance of a baseball glove (Supplementary Figs 7 and 8A). This is suggestive of a substrate-binding site given the compact, ball-like nature of its ferredoxin substrate.

As with other protein-binding TBDRs and in contrast to those which bind small molecules, the extracellular loop of the plug domain is highly extended, with residues 144 to 162 exposed to the external environment (Supplementary Fig. 8B). The neisserial transferrin receptor TbpA and the haemophore receptor HasR from *Serratia marcescens* utilize this loop for substrate binding and recognition[7,21]. While the extracellular loops display the

**Table 1 | FusA and ferredoxin crystallographic data collection and structural solution statistics.**

| | FusA native (4ZGV) | FusA PtCl₄ | *Arabidopsis* ferredoxin (4ZHO) | Potato ferredoxin (4ZHP) |
|---|---|---|---|---|
| *Data collection* | | | | |
| Space group | $P2_1$ | $P2_1$ | $P4_22_12$ | $I222$ |
| Cell dimensions | | | | |
| $a, b, c$ (Å) | 137.27, 79.89, 137.90 | 136.97, 78.44, 136.86 | 60.73, 60.73, 154.73 | 50.66, 71.69, 79.32 |
| $\alpha, \beta, \gamma$ (°) | 90.0, 90.7, 90.0 | 90.0, 90.56 90.0 | 90.0, 90.0, 90.0 | 90.0, 90.0, 90.0 |
| Peak wavelength | – | 0.88 | – | – |
| Resolution (Å)* | 48.95–3.20 (3.31–3.20) | 48.64–4.20 (4.54–4.20) | 60.73–2.34 (2.40–2.34) | 42.70–2.46 (2.52–2.46) |
| $R_{merge}$ | 30.5 (173.4) | 16.2 (58.8) | 6.6 (59.8) | 12.4 (59.6) |
| $R_{pim}$ | 13.7 (80.1) | 7.5 (26.3) | 2.7 (25.7) | 5.2 (24.9) |
| $I/\sigma(I)$ | 7.0 (1.3) | 9.9 (3.7) | 23.2 (4.2) | 13.9 (4.3) |
| $CC_{1/2}$ | 98.7 (56.7) | 99.7 (91.0) | NA | NA |
| Completeness (%) | 100.0 (100.0) | 99.9 (100.0) | 99.4 (98.4) | 99.7 (99.4) |
| Redundancy | 6.6 (6.4) | 6.7 (6.8) | 12.3 (11.8) | 12.3 (12.8) |
| | | | | |
| *Refinement statistics* | | | | |
| Resolution (Å) | 48.95–3.20 (3.31–3.20) | 48.64–4.20 (4.54–4.20) | 60.73–2.34 (2.40–2.34) | 42.70–2.46 (2.52–2.46) |
| No. of reflections | 49734 (4517) | 21,631 (4,438) | 12,894 (904) | 5,483 (382) |
| $R_{work}/R_{free}$ | 21.9/27.1 | | 19.51/23.79 | 17.75/21.68 |
| No. of atoms | | | | |
| Protein | 12,895 | | 1462 | 740 |
| Ligand/ions | (β-OG, LDAO) 129 | | (2Fe-2S cluster) 4 | (2Fe-2S cluster) 4 |
| Water | 0 | | 21 | 16 |
| r.m.s. deviations | | | | |
| Bond lengths (Å) | 0.011 | | 0.017 | 0.016 |
| Bond angles (°) | 1.66 | | 1.96 | 1.98 |

NA, not applicable; r.m.s., root mean squared.
Data from one crystal were collected for each structure.
See Supplementary Fig. 13 for representative electron density maps for each structure.
*Values in parentheses are for highest-resolution shell.

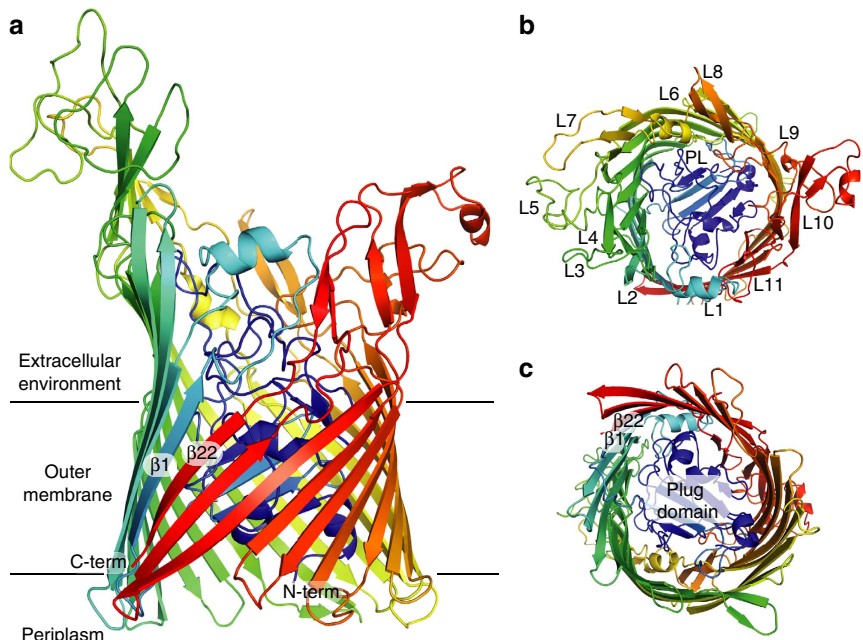

**Figure 2 | The crystal structure of FusA reveals a 22-stranded transmembrane β-barrel with glove-like ferredoxin-binding pocket.** (**a**) A cartoon representation of crystal structure of FusA, coloured as rainbow from blue at the N terminus to red at the C terminus. The transmembrane region and orientation in the outer membrane is inferred from homologous structures and detergent molecules resolved in structure. (**b**) View of FusA from the extracellular environment showing the position of the 11 extracellular loops (L1–L11) and the extended loop of the plug domain (PL). (**c**) View of FusA from the periplasmic space, showing the compact intracellular loops, the position of the plug domain and the location of the first and last strands of the β-barrel.

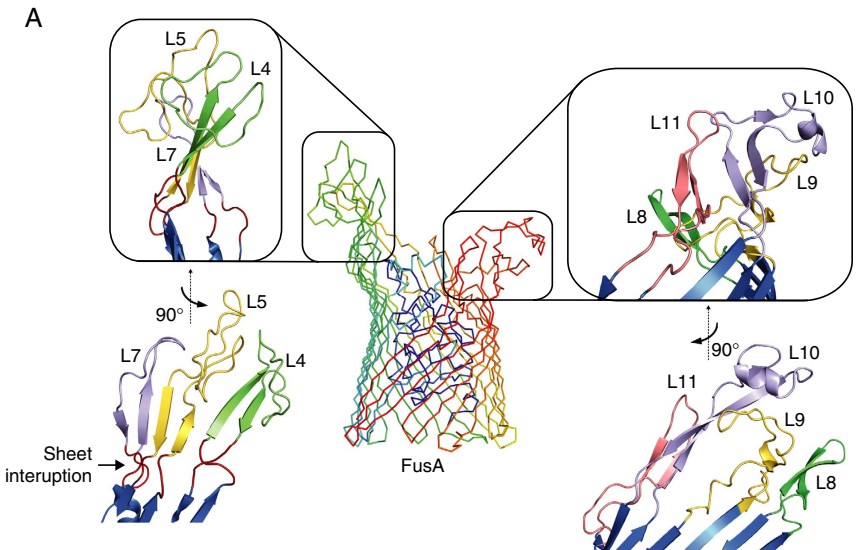

**Figure 3 | The extended outer loops of FusA form a glove-like ferredoxin-binding pocket through extensive inter-strand interactions.** (Centre) A wireframe representation of FusA with the large extracellular extensions formed by loops 4, 5 and 7 (left) and loops 8–11 (right) boxed. (Left) A zoomed cartoon representation of loops 4, 5 and 7 shows a β-wall structure ending in extended loops. The β-wall is separated from the β-strands of the FusA barrel by as serine-rich sheet interruption. (Right) A zoomed representation of loops 8–11 shows the extensive interactions between these strands creates a rigid structure composed of inter-loop β-hairpins, random coil and a single α-helix at the end of loop 10.

greatest sequence variation between FusA homologues, the charged surface this region presents is largely conserved (Supplementary Fig. 8C,D).

**FusA forms extensive interactions with ferredoxin**. To probe the interactions between FusA and its ferredoxin substrate, using NMR, we expressed and purified $^{15}$N-labelled, ferredoxin domain from pectocin M1 (PM1$_{fer}$), and ferredoxin isoform 2 from *Arabidopsis thaliana* (NP_176291) (Fer$_{ara}$). Titration of purified FusA into $^{15}$N-labelled Fer$_{ara}$ resulted in a decrease in the intensity of the $^{15}$N heteronuclear single quantum correlation (HSQC) spectral peaks that was proportional to FusA concentration, suggesting formation of a 1:1 complex in an intermediate to slow exchange regime (Supplementary Fig. 9). In contrast, titration of $^{15}$N-labelled PM1$_{fer}$ with FusA gave HSQC

spectra in which some peaks were relatively unperturbed while others became broadened and shifted with increasing FusA concentration. At a 1:1 stoichiometry, the majority of $PM1_{fer}$ peaks were still visible (Fig. 4a and Supplementary Fig. 10), indicating that formation of the complex between $PM1_{fer}$ and FusA occurs in the fast exchange regime. The FusA-binding surface of ferredoxin could therefore be mapped based on chemical shift perturbation (CSP) analysis. Using $^{13}C,^{15}N$-labelled $PM1_{fer}$ 56 backbone amides (of a total of 70 observed) were assigned, of which 34 display a CSP of >0.01 p.p.m. on addition of FusA (Fig. 4b and Supplementary Fig. 10). Residues displaying a CSP of >0.02 p.p.m. form a contiguous binding surface on the ferredoxin, predominantly localized to the ferredoxin β-sheet. This binding surface is composed of charged (K7, E14, D56, D64 and K86), polar (S54, Q67 and N80) and non-polar side chains (G11, L50, I51, Y70 and V84) (Fig. 4c).

**NMR-driven docking of ferredoxin to FusA.** Using the CSP data obtained from the FusA-$PM1_{fer}$ NMR experiments, we modelled the FusA–ferredoxin complex. We solved the crystal structures of *Arabidopsis* ferredoxin isoform 2 and potato (*Solanum tuberosum*) ferredoxin isoform 1 (CAC38395) ($Fer_{pot}$) and performed docking with extracellular portion of the FusA structure using the programme HADDOCK[22]. Docking solutions formed two major clusters in which the ferredoxins docked with the extracellular pocket of FusA, with the C terminus either

pointing up and away from the FusA barrel or at right angles to it (Supplementary Fig. 11A and Supplementary Data Set 1–4). The C-terminal up solution was favoured for both ferredoxins (Supplementary Table 2). Superimposition of the crystal structures of pectocin M2 with these solutions, based on their common ferredoxin domains, showed that only in the C-terminal up orientation was the cytotoxic domain of these structures accommodated without significant clashes with FusA, strongly suggesting it is the correct solution (Supplementary Fig. 11B). In this solution, the ferredoxin molecule is positioned directly over the pore of the FusA barrel and forms extensive interactions with the β-wall formed by loops 4, 5 and 7, and the plug domain loop of FusA (Fig. 5a,b). These interactions correlate well with the FusA-binding surface predicted by our NMR experiments, suggesting that these features are responsible for the initial stages of ferredoxin binding (Fig. 5c). When normal mode analysis was repeated on this docked complex, FusA underwent analogous conformational changes to unliganded FusA, indicating closure of the outer loops around the ferredoxin in a glove-like fashion (Supplementary Fig. 12 and Supplementary Movie 1). These conformational changes possibly represent those that occur upon ferredoxin binding and import by FusA.

**Growth enhancement of *Pectobacterium* by plant ferredoxin.** The FusA–ferredoxin docking simulations indicate that a discrete FusA-binding surface of ferredoxin mediates initial binding to the

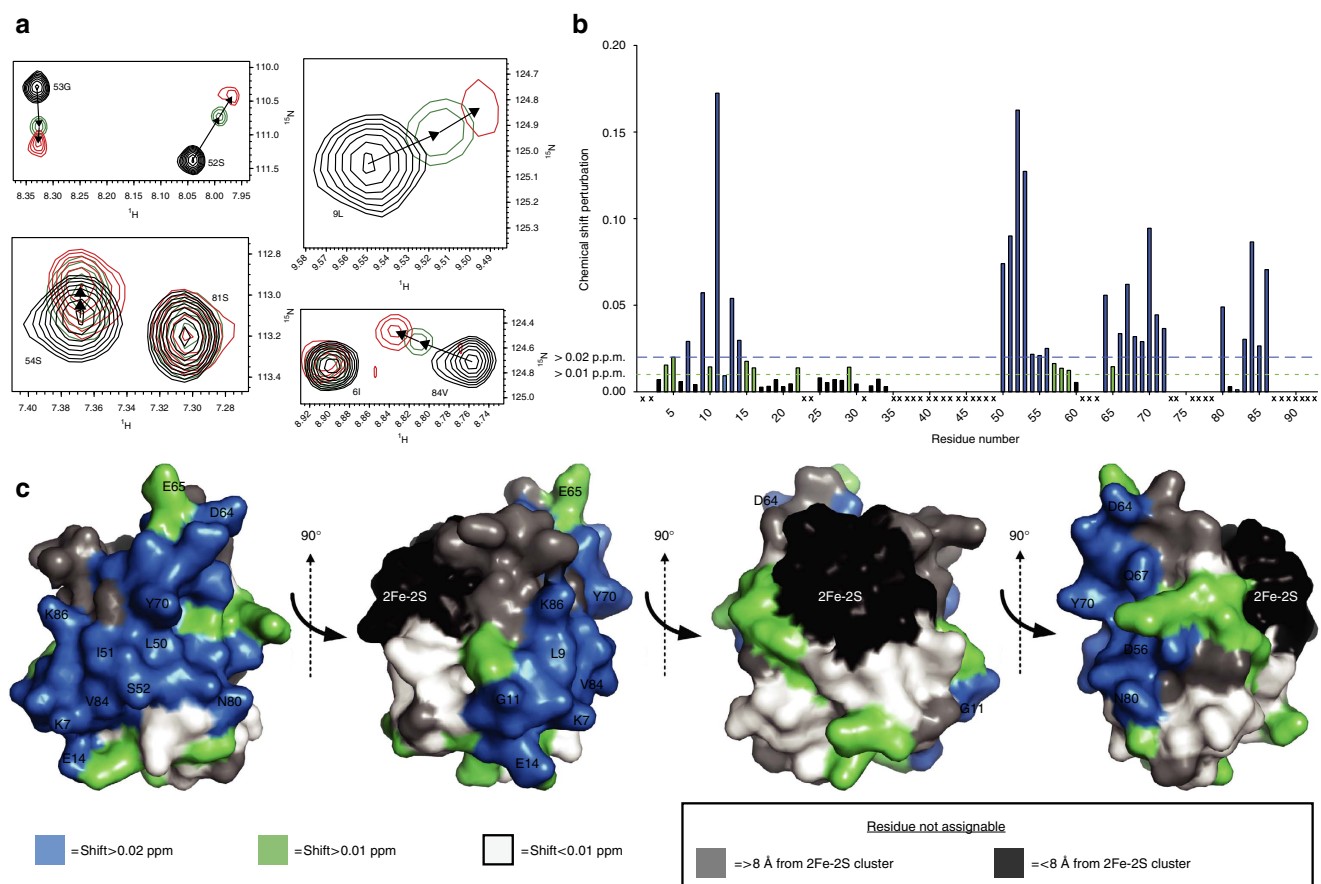

**Figure 4 | CSP analysis of the $PM1_{fer}$ HSQC spectra in the presence of FusA allows mapping of the FusA-binding surface. (a)** Representative chemical shifts of peaks from the $PM1_{fer}$ HSQC spectra on the addition of purified FusA. Black = 0:1 FusA to $PM1_{fer}$; green = 0.75:1 FusA to $PM1_{fer}$; red = 1:1 FusA to $PM1_{fer}$ (experiment performed three times). **(b)** A plot of CSP values for $PM1_{fer}$ residues, on the addition of FusA at a 1:1 molar ratio, residues with large CSP values represent residues likely form interactions with FusA. **(c)** Mapping of CSP data from **b** onto the surface to an atomic model of $PM1_{fer}$ shows residues, which experience a chemical shift >0.02 p.p.m. in the presence of FusA form a large connected binding surface on one side of the ferredoxin molecule.

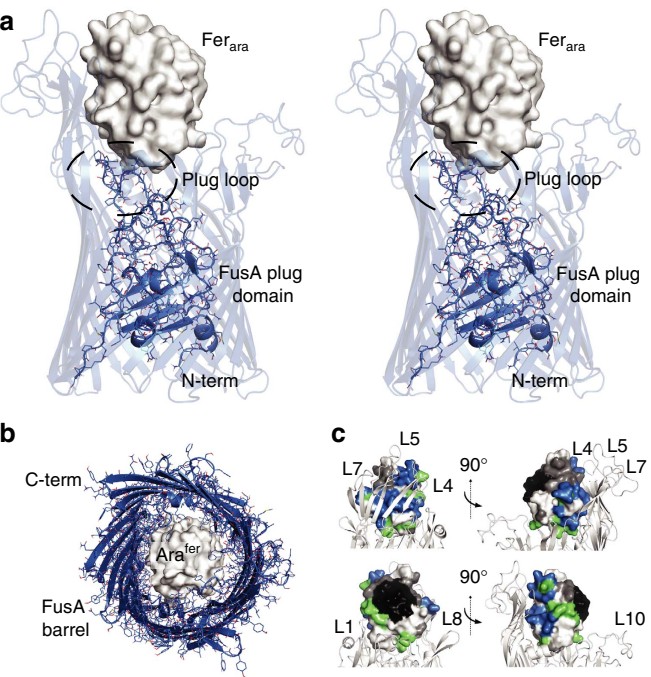

**Figure 5 | Ferredoxin forms extensive interactions with the FusA β-wall and plug loop and is positioned centrally in docking simulations.**
(**a**) Stereo view of Fer$_{ara}$–FusA docking solution, the FusA barrel is shown as transparent cartoon, while the plug domain is shown as a cartoon and stick model. Fer$_{ara}$ is shown as a van der Waals surface model in white. (**b**) The Fer$_{ara}$–FusA docking solution, the FusA barrel shown from the periplasmic side as a cartoon and stick model, with the plug domain removed, Fer$_{ara}$ is shown as in **a**. (**c**) van der Waals surface representation of PM1$_{fer}$ docked using HADDOCK to a cartoon representation of FusA. PM1$_{fer}$ is coloured as in Fig. 4c. The PM1$_{fer}$–FusA-binding surface (shown in blue) corresponds to the surface of PM1$_{fer}$ that interacts extensively with the β-wall formed by loops 4, 5 and 7 of FusA.

receptor. As such, it would be expected that sequence variation at this interface would affect the interaction of FusA with its ferredoxin substrate. To investigate this we tested purified leaf ferredoxins from potato isoform 1, maize (*Zea mays*) isoform 1 (ACG46956) and *Arabidopsis* isoform 2, which share 70–75% sequence identity, for their ability to enhance the growth of a panel of *Pectobacterium* isolates. Interestingly, these proteins enhanced growth to widely varying degrees (Supplementary Table 3 and Fig. 6a). *Arabidopsis* ferredoxin enhanced the growth of 14/17 strains, while potato ferredoxin enhanced 9/17 strains, with a noticeably weaker overall effect. Maize ferredoxin only enhanced one strain weakly, demonstrating that the discrete amino-acid sequence differences between the ferredoxins have a large effect on the ability of FusA to utilize the protein as a substrate (Fig. 6b,c). On the basis of our docking experiments, a number of the amino-acid differences between the three ferredoxins occur at the FusA–ferredoxin interface. Sequence divergence at the protein–substrate interface has been observed for the neisserial transferrin receptor TbpA and mammalian transferrin, where it has been shown to be driven by host–pathogen co-evolution[8].

## Discussion

Despite the universal importance of iron as a limiting nutrient for microbial growth and in pathogenesis, our understanding of the systems that bacteria use for obtaining it from their host is far from comprehensive[4]. This situation is even more pronounced for bacterial phytopathogens, where very little is known about the

role that specific iron import systems play in infection[1]. The discovery of FusA demonstrates that, like their mammalian pathogenic counterparts, phytopathogenic bacteria can use a TBDR to specifically target iron-containing host proteins[23]. The presence of the FusA in *Dickeya* and *Pantoea*, which are closely related to *Pectobacterium*, suggests the ferredoxin uptake system represents an important iron acquisition tool for soft rot pathogens[16]. FusA homologues however are not limited in their distribution to phytopathogens, with our analysis of microbial genomes revealing that more distantly related clades are present in proteobacterial species that adopt a commensal or pathogenic lifestyle with mammalian hosts. These species include *E. coli* and members of the genera *Shigella*, *Yersinia* and the family Pasteurellaceae[6,17]. Studies on FusA homologues in these species are limited, however upregulation in response to iron limitation and the presence of host factors has been demonstrated[24]. In a recent study, the *E. coli* homologues of both FusA and the predicted periplasmic protease FusC were shown to be important for the fitness of the UPEC strain CFT073 in systemic infection of mice. This study also showed that the genes encoding these proteins, designated *yddB* and *pqqL* belong to a single operon, which also contains *yddA*, a FusD homologue[25]. Interestingly, the order of the genes in this operon differs from that of the Fus operon, with the gene order varying further in homologues from *Proteus* and *Yersinia* (Fig. 1c). The maintenance of these genes in the same gene cluster, despite rearrangement, strongly suggests that they have a highly integrated function. While further investigation is required, together these data strongly suggest that the Fus proteins are members of a family that play an important role in the acquisition of iron from small iron-containing host proteins.

Our structural analysis of FusA shows that it is the member of a unique family of TBDRs. The extracellular and plug domain loops are highly extended, which is characteristic of a TBDR with a bulky protein substrate. However, the structure formed by these loops and the pattern of loop extension is distinct from other protein-binding TBDRs, showing that FusA has evolved to specifically bind its ferredoxin substrate[7,21]. Interestingly, sequence analysis of FusA homologues reveals the same pattern of loop extension, with the difference in length of homologues largely accounted for by the size of these extensions. Our modelling of the FusA–ferredoxin complex shows that the structures formed by these loops forms a large binding surface with the ferredoxin molecule and grasp it in a glove like fashion. It is tempting to speculate that this grasping motion of FusA, predicted by NMA simulations, represents the initial stages of substrate import *in vivo*. Whether the intact ferredoxin is then imported by FusA or somehow the iron sulphur cluster is liberated at the bacterial cell surface remains to be proven. However, the fact that the ferredoxin-containing pectocins are able to cross the outer membrane and enter the periplasm, through interaction with FusA suggests that importation of the intact ferredoxin is plausible[26]. Importation of the ferredoxin by FusA is also reasonable given then interior dimensions of the FusA barrel (Fig. 5b). This model would also provide a role for the predicted periplasmic protease FusC, which could cleave the ferredoxin on import, liberating the iron–sulphur cluster for import across the inner membrane by the ABC transporter FusD.

In summary this study reports and the structural characterization of a TBDR receptor that binds to iron–sulphur cluster containing ferredoxin domains.

## Methods

**Expression and purification of ferredoxins and pectocins.** The open reading frames (ORFs) for pectocin M1 and M2 and the plant ferredoxins (minus the stop codon) were cloned into pET21a (Invitrogen) and expressed in *E. coli* BL21 (DE3).

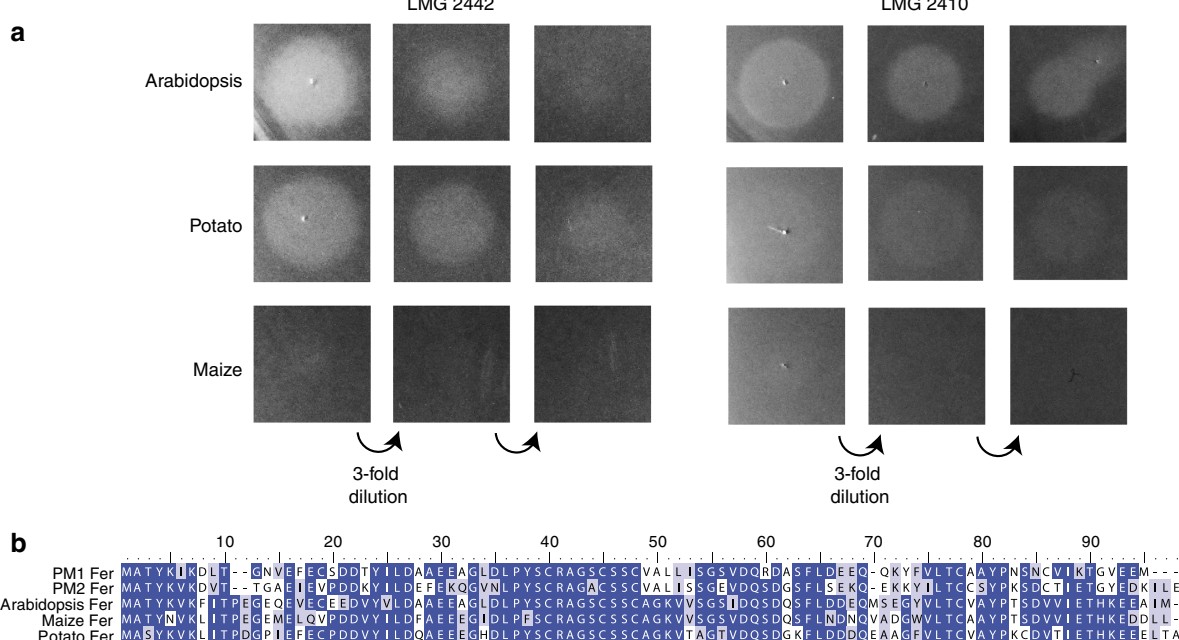

**Figure 6 | Plant ferredoxins with high levels of sequence homology enhance the growth of *Pectobacterium* species to greatly differing extents.**
(**a**) Growth enhancement of a soft agar overlay of *P. carotovorum* LMG2442 and LMG2410, on iron-limiting LB agar due to the application of threefold serially diluted purified plant ferredoxin at a starting concentration of 1 mg ml$^{-1}$. Ferredoxins from *Arabidopsis* and potato both enhance growth in these strains, whereas for maize ferredoxin no growth enhancement is observed. Refer to Supplementary Table 3 for a list of observed phenotypes from different *Pectobacterium* strains (experiment repeated >10 times). (**b**) Sequence alignment of *Arabidopsis*, potato, maize ferredoxins and the pectocin M1 and M2 ferredoxin domains showing the overall high level of amino-acid identity and the position of sequence differences between the homologues.

Cells were grown at 37 °C and protein expression was induced by the addition of 0.3 mM isopropyl-β-D-thiogalactoside (IPTG) at an OD$_{600}$ of ∼0.6. Cultures were grown for a further 6 h at 28 °C. Cells were collected and resuspended in 50 mM Tris-HCl, pH 7.9, 500 mM NaCl, 20 mM imidazole, 5% glycerol, 100 µg lysozyme, and Complete EDTA-free protease inhibitor cocktail tablets (Roche) were added. After disruption by sonication the supernatant was clarified by centrifugation and applied to a HisTrap-nickel agarose column (GE Healthcare) equilibrated in a buffer containing 50 mM Tris-HCl, pH 7.9, 500 mM NaCl, 20 mM imidazole and 5% glycerol. Bound protein was eluted with a linear gradient of 20–250 mM imidazole in lysis buffer. Pectocin-containing fractions were identified based on colour and analysis by SDS–PAGE. Pectocin-containing fractions were pooled and dialysed into 50 mM Tris-HCl, pH 7.5, 50 mM NaCl and purified using a Superdex S75 26/60 column (GE Healthcare) equilibrated in the same buffer.

**Cytotoxicity and growth enhancement assays.** The cytotoxicity of purified pectocins was tested using the soft agar overlay method[27]. A volume of 200 µl of mid-log phase culture of the test strain was added to 6 ml of 0.6% agar melted and cooled to 42 °C. The molten agar was then overlaid onto Lysogeny broth (LB) medium with or without 100–400 µM 2,2′-biyridine. Purified proteins (0.02– 10 mg ml$^{-1}$) were spotted directly onto the surface of the overlay, once solidified. Plates were incubated at 28 °C for 16 h, and monitored for zones of growth inhibition or enhancement.

**Identification of FusA.** A 2 l suspension culture of *P. atrosepticum* (*Pba*) LMG2386 was grown in LB with 200 µM bipyridine until stationary phase was reached. Cells were collected by centrifugation, resuspended in 20 ml of 50 mM Tris, 10 mM EDTA, pH 7.2. Protease inhibitors and 2 mg ml$^{-1}$ lysozyme were added and cells were lysed by sonication. Cellular debris were removed by centrifugation at 8,000*g* for 10 min and membranes were then pelleted by ultracentrifugation at 100,000*g* at 4 °C, for 1 h. Pelleted membranes were resuspended with a tight-fitting homogenizer in 20 ml of 50 mM Tris, 0.5% Sarkosyl (pH 7.2), left to solubilize for 20 min at room temperature, and the outer membrane fraction pelleted by centrifugation at 100,000*g* at 4 °C, for 1 h. The pellet was resuspended as before in 20 ml of 20 mM Tris, 1% *n*-octyl-β-D-glucoside (pH 7.2) and solubilized for 12 h at 4 °C. Purified pectocin M1 in 50 mM Tris, 50 mM NaCl, pH 7.5 was immobilized on a 1 ml His-trap column. Bound pectocin M1 was washed with 20 column volumes of Tris buffer (50 mM Tris, 500 mM NaCl and 10 mM imidazole, pH 8.0). The solubilized outer membrane (OM) fraction from *Pba* LMG 2386 was passed through the column, which was subsequently washed with 20 column volumes of Tris buffer as above. Proteins were eluted from the column with Tris buffer containing an increasing concentration of

imidazole (20, 50, 100 and 150 mM with pectocin M1 eluting at 100 mM imidazole). Control experiments were performed in which the *Pba* LMG 2386-solubilized OMs were passed down a column with no pectocin bound and imidazole elutions were undertaken for a column to which no OM fraction had been added. Proteins were visualized by Coomassie and silver staining of SDS–PAGE gels, and bands unique to the pectocin M1 plus OM fraction experiment were excised and identified by peptide mass fingerprinting.

**Cloning and expression of FusA.** To study FusA *in vitro*, homologues from a number of *Pectobacterium* strains (including LMG2386 and LMG2410) were screened for expression in *E. coli*. Despite extensive optimization, only the FusA homologue from the *P. atrosepticum* strain SCRI1043 (FusA$_{1043}$) yielded significant quantities of pure folded monodisperse protein (Supplementary Fig. 4) and as such was utilized for further studies. The method for achieving this expression was as follows: the FusA ORF from *Pba* SCRI1043 (FusA$_{1043}$) was amplified by PCR in its entirety or lacking the region coding for the 20 amino-acid N-terminal signal sequence (Δ20) (forward full length: GCATCCATATGAATAAGAACGTCTA TTTAATGATGC; forward Δ20: GCATCCATATGCAGCAAAATGATACCT CTGCCG; reverse: GCATCCTCGAGTTACCAGGTGTAAGCGACGC). The full-length ORF was ligated into pET21a at the NdeI and XhoI restriction sites. The Δ20 ORF was ligated into pET28a at the NdeI and XhoI restriction sites to encode a protein with an N-terminal His$_6$-tag and this construct was used to produce protein for structural and NMR studies. Plasmids were transformed into *E. coli* BL21 (DE3) for expression. Cells expressing full-length FusA$_{1043}$ were induced at OD$_{600}$ = 0.6 with 0.1 mM IPTG and grown at 28 °C for 12 h. The OM fraction from these cells was isolated, and co-elution experiments performed as for *Pba* LMG2386 membranes. Cells expressing FusA$_{1043}$ Δ20 as inclusion bodies were grown at 30 °C for 36 h in auto-inducing super broth[28]. Inclusion bodies were washed, refolded and purified based on the method described by Saleem *et. al.*[29]. Cells were collected and lysed as for pectocin purification, the insoluble fraction (containing the inclusion bodies) was isolated by centrifugation at 18,000*g* for 25 min and homogenized using a tight-fitting homogenizer in with 50 mM Tris, 1.5% N,N-dimethyldodecylamine N-oxide (LDAO), pH 7.5 and incubated at room temperature with shaking for 30 min. Inclusion bodies were pelleted by centrifugation at 18,000*g* for 25 min and homogenized once more in 50 mM Tris, 1.5% LDAO (v/v), pH 7.5, pelleted and washed once in 50 mM Tris (pH 7.5) before pelleting a final time. Inclusion bodies were then denatured in denaturing buffer (10 mM Tris, 1 mM EDTA, 8 M urea and 1 mM dithiothreitol (pH 7.5)) at a ratio of 0.5 g of inclusion body to 40 ml of buffer using a tight-fitting homogenizer, followed by incubation with shaking at 56 °C for 30 min. Insoluble material was then removed by centrifugation at 8,000*g* for 10 min.

FusA in denaturing buffer was then added drop wise to an equal volume of rapidly stirring refolding buffer (20 mM Tris, 1 M NaCl and 5% (v/v) LDAO (pH 7.9)), followed by stirring for 1.5 h. Refolded FusA was then dialysed (10–15,000 molecular weight cutoff membrane) into $2 \times 5$ l of dialysis buffer (20 mM Tris, 0.5 M NaCl and 0.1% LDAO (pH 7.9)) over 16 h at 4 °C. Refolded FusA was then purified by Ni-affinity chromatography, with dialysis buffer used for binding FusA to the nickel column and dialysis buffer with 0.5 M imidazole used to elute the protein. The protein was further purified using a superdex S200 26/60 column equilibrated in 50 mM Tris, 200 mM NaCl and 0.1% (v/v) LDAO (pH 7.9). For crystallization experiments in β-OG, FusA was exchanged into buffer containing 0.8–1% β-OG, by immobilization on a nickel column followed by washing with 10 column volumes of 50 mM Tris, 200 mM NaCl and 0.8–1% (v/v) β-OG (pH 7.9), and eluted with the same buffer with 0.5 M imidazole. The protein was then concentrated to 10–15 mg ml $^{-1}$ before dialysis (10–15,000 molecular weight cutoff membrane) for 20 h against 50 mM Tris, 200 mM NaCl and 0.8–1% (v/v) β-OG (pH 7.9), to remove imidazole and normalize β-OG concentration. Successful refolding of FusA was confirmed by analytical gel filtration and circular dichroism. Refolded FusA$_{1043}$ was concentrated and dialysed against a final buffer of 50 mM Tris, 50 mM NaCl and 0.1% LDAO for storage or 50 mM sodium phosphate and 0.1% LDAO for NMR experiments.

**Circular dichroism measurements.** Circular dichorism data were obtained using a Jasco J-810 spectropolarimeter (Jasco UK Ltd).

**Creation of ΔfusA strains.** The suicide pMRS101 plasmid was utilized in the creation of deletion mutants in *Pectobacterium* spp.[30,31]. The initial and final 1,000 bp of *fusA* gene were amplified by PCR and fused with a stop codon inserted at the fusion site. This cassette was inserted into pMRS101 at the unique SalI site and the ColE1 origin of replication excised by NotI digestion and re-ligation. After this step the vector (designated pKNFRKO2386) was propagated in *E. coli* SM10 λpir with 50 µg ml $^{-1}$ streptomycin as the selection agent. The vector was then transformed into the Pectobacterium knockout target (LMG2386) by electroporation and cells were selected on 50 µg ml $^{-1}$ streptomycin. It was found to be important to use 50 µg ml $^{-1}$ streptomycin rather than the published 25 µg ml $^{-1}$ streptomycin to prevent the isolation of spontaneously resistant colonies. Isolated streptomycin resistant colonies had the entire pKNFRKO plasmid recombined into the Pectobacterium genome via recombination in one 1,000 bp gene fragment. pMRS101 possesses the sacB gene, which imparts sucrose sensitivity on Gram-negative bacteria[30]. Recombination in the second 1,000 bp fragment, which leads to excision of the plasmid and either regeneration of the whole gene or creation of the knockout was selected for by growth in NaCl-free LB broth + 10% sucrose (LB sucrose), followed by serial dilutions onto LB sucrose agar (and LB streptomycin agar to test for completeness of plasmid excision). Sucrose resistant colonies were screened by PCR using primers for the target gene, with PCR products truncated to 2,000 bp indicative of a successful knockout. ΔfusA mutant and wild-type LMG2386 were tested for susceptibility to pectocin M1 using an agar overlay spot test as described above. A pectocin M1 susceptible phenotype was restored to ΔfusA by complementation with a plasmid containing a copy of *fusA* driven by a T5 promoter, spots tests were performed as above with the addition of 1 mM IPTG.

**Crystallization and structural solution of FusA.** Crystallization trials for FusA were conducted on refolded FusA$_{1043}$ in a buffer containing 50 mM Tris, 200 mM NaCl and 1% β-OG (pH 7.9), ∼800 conditions from commercial screens (Memplus, Memgold I/II, Morpheus, JCSG+, Midas, PACT, PGA and Memstart) were tested[32]. Very small needle-like crystals (5–20 µm) grew in a number of conditions, with the best forming in Midas condition H12 (15% PVP-K15, 25% PEG MME 5500 and 0.1 M Tris, pH 8.0). Crystals from the initial screen failed to diffract and so were subjected to extensive optimization. The results of this optimization yielded conditions (11–14% PVP, 14% PEG 2000 MME, 0.1 M Tris and 0.05 M MgCl$_2$ (pH 7.5), with a FusA concentration of 15 mg ml $^{-1}$ in 50 mM Tris, 200 mM NaCl, 0.8–1% (v/v) β-OG and 0.4% LDAO (pH 7.9)) producing crystals with somewhat improved morphology, still possessing a long needle-like structure, but being thicker and much larger (100–700 µm). Crystals were then looped, excess mother liquor removed, before cryocooling to 100 K and data collection at Diamond Light Source, Oxfordshire (DLS). The crystals were of the space group $P2_1$ and diffracted with extreme variability, with the best data recorded reflections to 3.0 Å (Supplementary Fig. 8A,B). Data were processed using XDS to yield a final data set with diffraction to 3.2 Å (refs 33,34). To obtain phases selenomethionine-labelled protein FusA was expressed in *E. coli* auxotrophic strain T7 crystal express (DE3) in M9 minimal media supplemented with selenomethionine, and refolded and purified as with unlabelled FusA$_{1043}$. However, selenomethionine-labelled protein failed to form well-diffracting crystals either in the Midas H12-optimized condition or in broad screens. As such, crystal drops were subsequently prepared as for native crystal and crystals were allowed to form, before finely powdered K$_2$PtCl$_4$ was added directly to the drop. Drops were resealed and allowed to equilibrate for 2–3 h before crystals were extracted as above. SAD data were collected at DLS at the platinum absorption edge wavelength of 0.88 Å. SAD data were processed using XDS with the resolution of best data set

extending to 4.2 Å (Supplementary Fig. 7A,B). Platinum sites were located using Shelx C/D with the best substructure solution consisting of eight sites (Supplementary Fig. 7C)[35]. These heavy atom sites and the experimental data set were provided to Autosol in the Phenix package for phasing and density modification, a clear solution was obtained and after many cycles of density modification using the Wang algorithm with electron density connected to two 22-stranded β-barrels apparent per ASU (Supplementary Fig. 7D). A poly-alanine model was constructed utilizing these experimental maps and used to phase the higher resolution native data by molecular replacement in Phaser[36,37]. The FusA model was then constructed using iterative cycles of model building in COOT and restrained (NCS and Prosmart secondary structure restraints) refinement in Refmac5 (Supplementary Fig. 8C)[38,39]. The β-barrel-specific secondary structure prediction programme BOCTOPUS proved highly accurate in predicting the position of transmembrane strands in the FusA sequence, which was invaluable in the initial stages of model construction[40]. Model quality was assessed using the Molprobity webserver[41]. Subsequent to the solving the structure of FusA, crystallization of FusA in complex with Fer$_{ara}$ and in complex with PM1$_{fer}$ was attempted. FusA at 15 mg ml $^{-1}$ in a buffer containing 50 mM Tris, 200 mM NaCl and 1% β-OG (pH 7.9) was mixed with an equal volume of the ferredoxin at 5 mg ml $^{-1}$ in 50 mM Tris and 200 mM NaCl (pH 7.9). Despite screening ∼800 conditions, no crystals were obtained.

**Crystallization and structural solution of ferredoxins.** Crystallization trials trails for purified C-terminal 6xHis-tagged *Arabidopsis* and potato ferredoxins (Fer$_{ara}$ and Fer$_{pot}$) were performed using a JCSG plus and PACT screens. Crystals formed in a number of conditions after ∼18 months. For Fer$_{ara}$ crystals from JCSG condition D6 (0.2 M MgCl$_2$, 0.1 M Tris and 20% PEG 8000, pH 8.5) were looped and cryoprotected by increasing the PEG 8000 concentration to 35% before cryocooling to 100 K in liquid nitrogen. For Fer$_{pot}$ crystals from PACT condition D5 (0.1 M Malic acid/MES/Tris and 25% PEG 1500, pH 8.0) were looped and cryocooled to 100 K in liquid nitrogen directly in mother liquid. Data were collected for both ferredoxin crystals at the Fe-K edge wavelength (1.74 Å) at DLS at 100 K. The best data sets diffracted to 2.34 and 2.46 Å for Fer$_{ara}$ and Fer$_{pot}$, respectively. Data sets could be solved both using Fe anomalous signal and by molecular replacement using the structure of spinach ferredoxin (Protein Data Bank ID: 1A70), using Autosol and Phaser, respectively[36,37]. The ferredoxins were built in COOT and refined using REFMAC5 (refs 38,42).

**Nuclear magnetic resonance.** Fast-HSQC spectra[43] were recorded for $^{15}$N-labelled *Arabidopsis* ferredoxin and pectocin M1 ferredoxin (100 µM) on a Bruker AVANCE 600 MHz spectrometer. The interaction of these proteins with FusA was explored by acquiring spectra for 1:1 ratio mixtures. LDAO (0.1 and 0.7%)-containing buffer was used as a negative control to ensure that chemical shifts were not caused by the presence of the zwitterionic detergent. In the case of the *Arabidopsis* ferredoxin–FusA complex, TROSY spectra were acquired to try to detect $^{15}$N-labelled ferredoxin in slow exchange with FusA. All spectra were processed with AZARA (W. Boucher, www.bio.cam.ac.uk/azara) and analysed with CCPNmr analysis[44]. A 300 µM stock of 15 N-labelled PM1$_{fer}$ or Fer$_{ara}$ was prepared in a buffer containing 0.1% LDAO and used as a reference of the CSP assays. Next, the 15 N-labelled protein was mixed with FusA at varying stoichiometry HSQC spectra were recorded. For sequence-specific backbone assignment all resonances were measured on $^{15}$N-labelled PM1$_{fer}$ (600 µM) and $^{15}$N-$^{13}$C PM1$_{fer}$ (350 µM) in 100 mM sodium phosphate, 5% D$_2$O (pH 6.9). 3D $^1$H-$^{15}$N NOESY (mixing time 100 ms) and 3D $^1$H-$^{15}$N TOCSY (mixing time 60 ms) spectra were acquired on the $^{15}$N-labelled sample, and 3D HNCO, HNCACO, HNCACBCO and HNCACB were acquired on the triple labelled $^{15}$N-$^{13}$C PM1$_{fer}$ sample. Spectra were processed with AZARA (W. Boucher, www.bio.cam.ac.uk/azara) using the MaxEnt method), and backbone assignment was carried out with the CCPNmr software package.

**Analytical ultracentrifugation.** Sedimentation velocity was carried out in a Beckman Coulter Optima XL-I analytical ultracentrifuge using an An-50 Ti 4-hole rotor. FusA (90 µl) at concentrations ranging from 0.2 to 10 mg ml $^{-1}$ was loaded into a 3 mm path-length centrepiece and spun at 49,000 r.p.m. for ∼12 h at 15 °C. Scans were collected every 7 min using absorbance optics (at 280 nm; a radial range of 5.8–7.2 cm, and radial step-size of 0.005 cm). The buffer used was 50 mM Tris, 200 mM NaCl and 2 mM α-sulpho myristic acid (C14SF), pH 7.5, and this buffer lacking C14SF as a reference. Data were analysed with SEDFIT[45] using the continuous c(s) distribution model. SEDNTERP was used to calculate the partial specific volume, the buffer density and viscosity at 15 and 20 °C.

**HADDOCK docking.** Docking was performed using the HADDOCK server http://haddock.science.uu.nl/services/HADDOCK2.2/ (ref. 22). For docking of the PM1$_{fer}$ domain an atomic model was generated using modeller[46]; this model along with the *Arabidopsis* and potato ferredoxin crystal structures were refined using water refinement in HADDOCK, and an ensemble of 10 conformations generated, before docking experiments. For the initial round of docking all extracellular loop residues from FusA were defined as passive interactors. Docking from this round led two populations of solutions, one originating from ferredoxin docking into the

extracellular pocket of FusA and the other from spurious interactions with the outside of the FusA barrel. As such in subsequent rounds the inside of the extracellular loops of FusA were defined as passively interacting with the ferredoxins. There residues on FusA were 100, 101, 102, 107, 101, 144, 145, 146, 147, 148, 149, 150, 151, 153, 156, 158, 159, 162, 163, 234, 236, 239, 243, 327, 376, 378, 380, 382, 383, 384, 385, 396, 397, 398, 399, 400, 401, 403, 405, 461, 462, 463, 464, 465, 466, 467, 468, 469, 470, 471, 472, 473, 474, 492, 495, 497, 585, 587, 589, 590, 592, 594, 595, 596, 597, 598, 599, 601, 602, 603, 604, 605, 606, 648, 650, 652, 653, 654, 655, 656, 658, 705, 706, 710, 712, 713, 715, 726, 727, 732, 770, 771, 772, 773, 774, 775, 776, 778, 779, 790, 791, 792, 793, 795, 796, 837, 839, 841, 843, 844, 845, 846, 847, 848, 849, 850, 851 and 852. For PM1 and PM2 the following residues were define as active (based on average NMR CSP $+$ 1 s.d.): 9, 50, 51, 52, 54, 70, 84 and 86. For potato and *Arabidopsis* the same active residues were defined (with amino-acid numbering shifts to take account insertions into these proteins): 9, 52, 53, 54, 56, 73, 87 and 89. A 25% random removal of restraints was applied, the N termini for all proteins was set as charged, C termini for all proteins was set as uncharged. See Supplementary Table 3 for statistics for the top clusters from the docking runs.

**NMA analysis.** NMA analysis was performed using the *elNémo* server: http://www.sciences.univ-nantes.fr/elnemo/[47]. The docked structure of FusA and PM1$_{fer}$ from HADDOCK was submitted for analysis in five modes, with DQ min $= -300$, DQ max $= 300$ and DQ step $= 20$. Outputs from the different modes where manually screen for plausible movement of the molecules.

**Phylogenetic analysis of FusA homologues.** Sequence identity searches to identify FusA homologues were performed using the HMMER (phmmer) algorithm[48] against the Uniprot representative sequences (rp75) and NCBI references sequence (RefSeq) databases, E value cutoffs of 1e-40 and 1e-50 were applied for the rp75 and RefSeq databases, respectively. Sequences identified in these searches were output in FASTA format and manually curated to remove multiple instances of identical sequences from the same species and truncated sequences (<500 amino acids). This yielded 45 and 219 sequences for the rp75 and RefSeq databases, respectively. To determine evolutionary relationships between the FusA homologues these sequences were analysed using CLANS (CLuster ANalysis of Sequences)[49] with a P value cutoff of 90, network-based cluster analysis was then performed to identity clusters. For the rp75 data set the genomic context of FusA was analysed for the presence of sequences homologous of the putative FusA operon.

**Data availability.** All crystallographic coordinates and associated structure factors produced in this study are available in the Protein Data Bank with the following accession codes: FusA $=$ 4ZGV; *Arabidopsis* ferredoxin $=$ 4ZHO; and potato ferredoxin $=$ 4ZHP. DNA/protein sequences generated in this study have been deposited in the GenBank Database with the accession codes FusA from *P. atrosepticum* LMG2386 $=$ KX258448 and FusA from *P. carotovorum* LMG2410 $=$ KX258449. The Rp75 reference proteome database used for identification of FusA homologues is available from http://www.uniprot.org/proteomes/. All other data to support the findings of this study are either available with this article (coordinates from HADDOCK docking run) or are available from the corresponding authors on request.

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

## Acknowledgements

We thank the Diamond Light Source for access to beamlines I02, I03, I04, I04-1 and I24 (proposal numbers MX6638 and MX8659). The work was funded by the BBSRC (BB/L02022X/1), a Kelvin-Smith Scholarship from the University of Glasgow and a Sir Henry Wellcome Fellowship awarded to R.G.: award number 106077/Z/14/Z. During this work I.J. was supported by a studentship from the Wellcome Trust: award number 093592/Z/10/Z.

## Author contributions

Conceived and designed the experiments: D.W., R.G., I.J. and A.M.J.J.B.; performed the experiments: R.G., I.J., K.M. and A.M.J.J.B.; analysed the data: D.W., R.G., I.J., A.M.J.J.B., S.K., B.S., A.W.R. and O.B.; contributed reagents/materials/analysis tools: D.W., S.K., B.S., R.J.C.; wrote the paper: D.W., R.G., I.J., A.M.J.J.B., A.W.R., R.J.C., J.M., O.B. and B.S.

## Additional information

**Competing financial interests:** The authors declare no competing financial interests.

