## [Peer Review File · Nature Communications]

Reviewer #1 (Remarks to the Author):

Key results

The authors previously characterized the activity of a novel type of bacteriocin, represented by pectocins M1 and pectocin M2 that are produced by *Pectobacterium* strains to kill some other pectobacteria. These protein sequences share a similar bi-domain composition not previously described in bacteriocins. By solving the structure of pectocin M2 they subsequently demonstrated that the cytotoxic domain is linked to a globular ferredoxin domain that displays striking sequence/structural similarity to [2Fe-2S] ferredoxins produced by plants.

In the current manuscript they describe the identification and structure elucidation of the outer membrane protein that is targeted by pectocin M1 as its receptor on the surface of susceptible cells. As pectobacteria are major phytopathogens causing soft rot on a wide range of plant hosts, the authors examine the plausible hypothesis that this receptor's natural function is to take up ferredoxins from their host plant as a source of iron. Such 'iron piracy' occurs by mammalian pathogens but this is not yet documented for phytopathogenic bacteria. As suggested by the title, the authors here claim to demonstrate that such iron piracy by *Pectobacterium* is mediated by the same type of receptor that is parasitized by pectocins to gain entry in susceptible cells.

Originality and interest

Although other TonB-dependent receptors (TBDR) are known to be involved in uptake of certain bacteriocins (e.g. colicins of *Escherichia coli* and pyocins of *Pseudomonas aeruginosa*) and some of these mediate iron uptake through siderophores, this work on *Pectobacterium* bacteriocins reveals yet another variant of the strategy to parasitize iron metabolism components by the identification of a novel type small-protein-binding TBDR with the capacity to scavenge (plant) ferredoxins. The characterization of this TBDR has broader implications as sequence-related TBDRs are not restricted to phytopathogenic relatives of pectobacteria but also occur in several mammalian pathogens where their function or structure is not yet known.

Specific comments regarding methodology, results, and conclusions

To probe the interaction of the TBDR with its presumed substrates by NMR, the pectocin M1 domain and selected plant ferredoxins were produced recombinantly. Additional information about the interaction interface was obtained by docking analyses, after first elucidating the structure of two of the plant ferredoxins used. These experiments support the predicted substrate recognition (although not quantitatively with respect to affinity) and provides valuable information on the mode of binding that can be subject to further scrutiny in follow-up work, for instance by analyzing mutant forms affected in the binding patches. In these experiments, recombinant Arabidopsis and potato ferredoxins were used, but this choice is not really motivated (spinach ferredoxin I was used in previously published competition experiments; ref. 15). The TBDR characterized is from a potato isolate but, typically, pectobacteria do not have a preferred host and are characterized by a broad host spectrum. Moreover, a particular plant species typically produces different ferredoxins that differ in amino acid sequence. In the manuscript, no information is provided on which ferredoxin isoforms (chloroplast, root, tuber, ...?) were used (and why). It could be informative to mention whether the authors tried/are trying to obtain co-crystals with the isolated pectocin domain or with a plant ferredoxin.

For most of the paper, it is well written and the data are clearly presented (some minor suggestions for improvement are listed below). However, the last part of the Results section of the manuscript that describes data in support of the claim of iron piracy through plant ferredoxin uptake by this TBDR, is not of the same high level as the previous experiments and not convincing as a validation of the proposed TBDR-mediated iron uptake. For a panel of *Pectobacterium* strains, it was found that growth was promoted, to variable extents (only roughly scored as weak/medium/strong, depending on the strain), in iron-limiting medium supplemented with plant ferredoxins (three different types, also differing in their growth enhancement capacities). This observation is not really novel since such strain-dependent growth enhancement was described previously by the authors, albeit with spinach ferredoxin (ref. 15). The Methods section refers to a concentration range of iron chelator that was used to vary iron availability (L356) but it is not clear to which condition the data in Table S4 actually correspond. Although this experiment indicates that plant ferredoxin-derived iron becomes available to (some of) the pectobacteria, it does not unequivocally prove that TBDR-ferredoxin-mediated iron piracy is involved. For instance, can it be excluded that the ferredoxins are first degraded and the released iron would be taken up by the conventional (siderophore-dependent) route? As described earlier for spinach ferredoxin (ref. 15), the observed outcome ranges from lack of growth to well supported growth of individual strains, which may be expected due to the sequence divergence between plant ferredoxins. It is not clear what conclusive information regarding the proposed iron piracy strategy can be gained by such growth enhancement assay with multiple strains. For (most of) these strains, the TBDR sequence of interest is not known and, hence, sequence variation at the TBDR interface cannot be mapped on them to find possible correlations with corresponding sequence variation localized in the ferredoxins. As such, Figure 6 does not add much to the manuscript by mapping sequence divergence in three plant ferredoxins and the ferredoxin domain of pectocin M1 (there is some redundancy with ferredoxin comparison in Fig. 5D).

If the TBDR specifically supports the acquisition from plant ferredoxin-carried iron, one expects that this acquisition would be abolished in a TBDR-lacking mutant (concomitantly to becoming immune to the pectocin). Such immunity was demonstrated in this work, using the TBDR-mutant constructed

for strain LMG 2386, but surprisingly, this mutant was not included in the growth enhancement experiment to compare its phenotype with the parent strain. Have the authors also tried to compare growth enhancement of strain LMG 2386 and its TBDR-deficient mutant by supplementation with the recombinant ferredoxin domain of pectocin M1 (should be a good interaction partner and could be a potential iron source)? Notably, growth enhancement of this strain (initially used to fish out the pectocin M1 target) and also of the strain for TBDR structure elucidation (SCRI1043) is only weak and evident for only one/two of the three ferredoxins tested. Could it be that the identified TBDR prefers other (plant) ferredoxins?

Demonstration of abolished or strongly diminished acquisition of iron (isotope form) from iron-loaded ferredoxin, externally supplied to the TBDR mutant cells, would be more convincing than the observed growth enhancements.

Apparently, there is also strong sequence divergence between the ferredoxin domains of the pectocins M1 and M2. It would be informative to include the pectocin M2 domain in sequence comparisons. Although most experimental work refers to pectocin M1 and its domain, the available crystal structure of pectocin M2 is introduced in the Results section on docking (L176). It should be made clear to the reader that pectocin M2 is not just an orthologue of pectocin M1 but is quite different from pectocin M1. The M1 and M2 ferredoxin-like parts share only about 60% amino acid sequence identity, which is in the same range of sequence divergence as between the M1 domain and the three plant ferredoxin used. Such differences likely have an impact on the respective interaction interfaces. Have the authors attempted to isolate the receptor for pectocin M2 (with crystal structure known) using the same approach that was successful for pectocin M1?

In the Discussion, the authors claim that by the discovery of FusA they have demonstrated for the first time that phytopathogenic bacteria are capable of obtaining iron from host proteins (L210-213; L249-251). Although such iron piracy is likely to occur, based on the results presented here, it has not been unequivocally demonstrated. Therefore these statements should be toned down and the title should be modified to reflect this.

The distinction between the (plant) ferredoxin protomers and the ferredoxin domain that is part of the pectocin M proteins (fused to an extra killing domain) is often not clear from the description and nomenclature used. The latter is also designated as a 'ferredoxin' whereas it is actually a ferredoxin-like domain (even when recombinantly expressed as a separate polypeptide it should still be clear to the reader that this is derived from a bi-domain protein). This ambiguity in the descriptions is maintained in the titles: L87, L117, 'ferredoxin receptor' refers actually to the pectocin M1 receptor (this may give the false impression that already at that stage it has been proven that the receptor functions in binding/uptake of plant ferredoxins); L148, 'globular ferredoxin' is used for both plant protein and recombinant bacteriocin domain. It may be helpful for the reader to distinguish both by adopting different abbreviations for the ferredoxin domains and the 'genuine' ferredoxins. For instance: keep PM1(fer), with the subscript specifying the domain; instead of the Plant(fer)

designation Fer(plant) may be used, with the subscript referring to the species origin (preferably derived from the Latin species name). Also note that apparently two different designations are used for the domain of pectocin M1 (L483, L486, Figure 4 also refer to it as MPCl(fer), but this is nowhere defined in the text).

The bioinformatic analysis shows that FusA represents a novel type of TBDR that is also found in several mammalian pathogens. Several additional gamma-proteobacteria are mentioned in the text but it is not highlighted that the analysis also identifies some important pathogenic beta-proteobacteria (like *Neisseria*, *Bordetella*) and epsilon/delta-proteobacteria (*Campylobacter*). The FusA-related proteins are collectively described as homologues but it should be pointed out that the sequence identities of many of these proteins is low (for most representatives listed in Table S1 sequence identity is less than 25%). As such, the nature of the substrates may also be quite different. The conclusion drawn from this analysis (L113-115 of Results) is rather speculative and more appropriate for the Discussion section.

Minor remarks

L66-69. Strictly speaking, bacteriocins like colicins, pectocins, ... are not only used for intraspecies competition (acting against strains of the same species), since examples are known (including pectocins studied here) where producer and target belong to different species (but usually belonging to the same genus).

When ferredoxin sequences are used for a particular plant species the corresponding database accession number /unique identifier should be provided (presence of isoforms).

Figure S1. Mark lanes with numbers. Indicate the position of pectocin M1. Use same sequence identifier as in the text (PCC21_007820); Pba has not been defined. Correct: cartovorum.

Figure S2. Represents data from a smaller (A) and a larger (B) subset and data in panels A and B thereby overlap. Cluster numbers used in panel A are not indicated in panel B. Why not use only the larger subset to avoid redundancy (B is said to "reinforce" A) and improve clarity of this figure?

Figure S6. The codes (FusA_1043 etc) used should be clarified in the text by indication of the source (species names/full strain numbers) of these FusA sequences in the legend

Figure S8. Enlarge panels A and B for better visibility. Panel C: is based on comparison of homologues in Figure S2, but the actual sequences used are not specified (those of small or large subset?, only those of a particular cluster including *Pectobacterium*?)

Table S1. No representative of *Neisseria* included?

Table S4. Rephrase sentence of legend: "... a qualitative measures the strength of the zone ..."
(meaning not clear). Are data about the susceptibility of these strains to pectocin M1 and pectocin M2 available? The effect is shown only for one strain at a single (high) concentration of the three plant ferredoxins (as an example in Figure 6.A). A better view of the concentration-dependent capacity to enhance growth would be provided by showing the extent of growth obtained with the different concentrations tested (dilution series). This could be presented as a multi-panel figure illustrating concentration-dependent growth profiles for these strains (or for representative strains with a particular profile).

L387. Was the FusA protein expressed without the first 20 amino acids actually used in some of the experiments?

The correct designations for proteins (FusA, ...; non-italic) and genes (*fusA*, ...; italic) should be used throughout the text and in figures.

It is important to always specify the strain that is used (not only in Methods but also in Results). For instance: the *P. atrosepticum* strain used for *fusA*-mutant construction (L96-98) is not mentioned; the strain from which the *fusA* gene was used for structural characterization is described without species designation as '*Pectobacterium* SCRI1043' (L119; L383-384). Abbreviations used for bacterial (sub)species are not really standard (for instance 'Pba, 'Pbc'). They should be defined when the full name is used for the first time.

Reviewer #2 (Remarks to the Author):

Manuscript NCOMMS-16-01698-T Review

This manuscript reports the X-ray crystallographic structure of the phytopathogenic TonB-dependent receptor FusA along with convincing NMR data that demonstrate its ferredoxin-binding properties. The NMR data are used as constraints to obtain a model of a ferredoxin-FusA complex by automated docking. The structure of this protein is an interesting departure from those of most TonB-dependent receptors solved to date, since it has extensive extracellular loops with local folding that presumably form a protein-protein interaction interface with ferredoxin ligand. The authors further perform an analysis of phylogenetically related ferredoxin proteins to identify likely FusA-binding residues. Although the authors do not go as far as to demonstrate ferredoxin transport through FusA, they propose this as a plausible molecular mechanism given the dimensions of the FusA barrel domain.

The science in this manuscript is solid, and the authors have performed thorough and carefully designed experiments to support their hypotheses. This manuscript represents a significant advance in our understanding of TonB-dependent receptors, both in terms of FusA being the first TBDR to be identified as ferredoxin-binding as well as being the first structure of a protein-binding TBDR with extensive extracellular regions that comprise a protein interaction interface. Although a FusA-ferredoxin co-crystal structure would have been desirable, the data-constrained docking provides an useful model in the absence of the co-crystal structure. This manuscript thus deserves to be considered for publication in Nature Communications following some revisions.

Listed below are major and minor points that the authors should address.

Major points:

1. Given the fast exchange NMR data exhibited for PM1fer, is not possible to obtain an estimated KD of interaction between FusA and this ligand via a saturation experiment? Such information on the affinity of the complex would be useful. Also, the fact that ¹⁵N-labelled ARAfer exhibits slow exchange while ¹⁵N-labelled PM1fer exhibits fast exchange suggests a significant difference in binding affinity. Even if saturation experiments are not possible, perhaps the authors should comment further on this apparent affinity difference in the manuscript. Why does PM1fer bind with apparent lower affinity?

2. Lines 229-230: The authors claim that the Fus proteins "represent the first members of a family that plays an important role in the acquisition of iron from small iron-containing host proteins". This statement should be slightly re-written for clarity, perhaps changing 'small iron-containing host proteins' to 'ferredoxin-containing proteins'. There are other TBDRs in nature that bind to iron-loaded protein ligands at their extracellular loops. The TBDR HgbA from Haemophilus and Actinobacillus, for example, is a hemoglobin-binding protein that is predicted to have extensive extracellular loop regions forming a protein interaction interface to bind hemoglobin for iron extraction. In fact, the authors need to discuss FusA in the context of HgbA, since to my knowledge this is the only other TBDR known to bind to another protein as a ligand.

3. The authors describe FusA as a TonB-dependent receptor, yet go on to state that it interacts with the 'TonB-like' protein FusB. Just how similar is FusB in comparison with TonB? For example, from what I can see of the primary sequence (Fig. S6), FusA has a Ton Box sequence (53DTILV57, not resolved in the electron density). Is this also a FusB-binding sequence? The authors should elaborate on this. Does Pectobacterium also express TonB in addition to FusB?

4. Figs S4, C&D: The electron densities shown in these figures appear quite smoothly distributed given the resolution. Are these 2Fo-Fc maps? The nature of the maps should be specified in the figure legend. Ideally, something like a simulated annealing omit map should be shown as a better assessment of quality of the diffraction data.

5. Figure 5. This is a very busy figure with a lot going on in it. The stereodiagrams are too small to effectively portray anything of interest in 3D. The figure that could really benefit from a stereoview is Fig. 5A. In contrast, Fig 3B is required to show the overall barrel cross section in comparison with the diameter of the docked ferredoxin -- a 3D representation is not necessary for that. Furthermore, Fig. 5D uses the coloring scheme employed in Fig. 6 to indicate positions of sequence dissimilarity between related proteins -- it seems out of place in Fig. 5. I would suggest moving Fig. 5D into Fig. 6 (perhaps as Fig. 6D), and to convert Fig. 5B to a 2D figure. This would free up space to enlarge Fig. 5A

Minor points:

1. Line 249-250: "TBDR receptor" - the use of 'receptor' is redundant here.

2. Fig 5.: the ferredoxin ligand is shown as a surface representation, which is effective; however, is this a van der Waals surface? It looks like one. The ligand should instead be rendered as a solvent-accessible surface in Fig. 5 to more accurately indicate the dimensions of the ligand relative to the barrel pore.

3. Fig. S5 C&D: Extracellular loops around M732 are clearly being stabilized by an NCS-related protomer in the asymmetric unit. What's stabilizing the other loops (i.e. the other half of the "catcher's mitt"? Crystal contacts?

Reviewer #3 (Remarks to the Author):

Summary.

Iron is essential for life and essential for the pathogenesis of most bacterial pathogens. Since free iron is rare, Gram-negative bacterial pathogens have evolved ways of acquiring iron from host proteins such as transferrin. They do so using a system that depends on a receptor/transporter that sits in the outer membrane and an energy transduction complex that sits in the inner membrane that is powered by the proton motive force. The receptor/transporter is responsible for binding iron-containing proteins and transporting either iron or the iron-sideophore complex across the outer membrane into the periplasm. Once in the periplasm, the iron is then further shuttled into the cytoplasm by other proteins in the pathway. Here, the authors present the structure of FusA, which is the first structure identified that directly targets ferredoxin as an iron source. The authors identify FusA using a fishing experiments and later determine its structure using X-ray crystallography. Subsequent studies identify it as a ferredoxin binding protein that is important for plant pathogens. The authors also determine the structures of ferredoxins and modeling the binding between the ferredoxin and the receptor, based on NMR and other studies.

Comments/suggestions (in no particular order):

1. Overall, well written and organized paper. A number of grammar errors but I am sure those will be fixed before being finalized.
2. The manuscript is short and concise, especially given there are many figures (including supp figs), but does a nice job of following up on the structural studies, etc.
3. In the pull down assay that discovered FusA, the bait (pectocin M1) is in far excess, yet 2 liters of a saturated culture was used to yield only a barely detectable band on an SDS-PAGE gel. The affinity for such interactions has historically be very high, so what does this say about the expression

levels of FusA? Or the affinity? Did the authors ever develop an antibody against FusA to determine what percentage of the FusA they were actually capturing in this assay?

4. The authors never show they can actually make a FusA:ferredoxin complex, nor do they indicate the possible binding affinities for one another? Therefore, for the NMR experiments, we are left to assume this, but I think some information here is needed since it hasn't technically be established in vitro using the refolded sample. Also, admittedly not an NMR expert, it seems odd to me that with increasing FusA, the peaks disappear almost entirely? What does this say about the fold of ferredoxin in the presence of FusA? What is happening to the structure of ferredoxin here to make the peaks disappear? If I had looked at the 1:1 complex spectra first, I would have concluded a lack of structure in ferredoxin when bound to FusA or that FusA is somehow unfolding ferredoxin. Can the authors clarify this?

5. I strongly suggest that the authors make the models of their protein complexes available as supplementary download so that others in the field can properly assess and utilize them as needed.

6. The authors did not follow up their modeling studies of the complex with some mutagenesis to determine if they can disrupt the binding or not. This would make the manuscript much better and more conclusive.

7. The crystallographic table is missing some info; 'Mean/Wilson plot B-value'

a. Not necessary to put down percentiles for MOLPROBIT scores, those are relative and will not remain constant.

b. Ramachandran scores do not add up to 100% for potato ferredoxin

c. In footnote c, what does 'sup 16' indicate?

8. For the movie, it would be nice to include a title slide. Also, I suggest looping the morph such that it not only closes, but opens and maybe cycles a few times, and maybe even show this from a few different angles.

Reviewer #1

1. To probe the interaction of the TBDR with its presumed substrates by NMR, the pectocin M1 domain and selected plant ferredoxins were produced recombinantly. Additional information about the interaction interface was obtained by docking analyses, after first elucidating the structure of two of the plant ferredoxins used. These experiments support the predicted substrate recognition (although not quantitatively with respect to affinity) and provides valuable information on the mode of binding that can be subject to further scrutiny in follow-up work, for instance by analyzing mutant forms affected in the binding patches. In these experiments, recombinant Arabidopsis and potato ferredoxins were used, but this choice is not really motivated (spinach ferredoxin I was used in previously published competition experiments; ref. 15). The TBDR characterized is from a potato isolate but, typically, pectobacteria do not have a preferred host and are characterized by a broad host spectrum.

In our previously published work (Ref 15) where we reported the discovery of the ferredoxin containing bacteriocins and the subsequent finding that *Pectobacterium* spp. could obtain iron from plant ferredoxin, spinach ferredoxin was used and obtained commercially. However, in the current work proteins were recombinantly expressed. For this study we chose 3 ferredoxins, from distantly related plants (Arabidopsis, potato and maize) and in this respect our motivation was to begin to determine if there is any specificity with respect to the ability of *Pectobacterium* spp. to obtain iron from ferredoxins from different plant species. Our results clearly demonstrate that there is a difference (we expand on this in the reply to Point 4)

2. Moreover, a particular plant species typically produces different ferredoxins that differ in amino acid sequence. In the manuscript, no information is provided on which ferredoxin isoforms (chloroplast, root, tuber, ...?) were used (and why).

We have included information on which ferredoxins isoforms were used in the manuscript. All ferredoxins utilised were leaf type ferredoxins (Arabidopsis = Isoform 2, Maize = Isoform 1, Potato = Isoform 1).

3. It could be informative to mention whether the authors tried/are trying to obtain co-crystals with the isolated pectocin domain or with a plant ferredoxin.

We did attempt crystallisation of the Fusa-ferredoxin complex, however no crystals were obtained, we appreciate that this information could be useful however and have included the following text in the online methods 'Subsequent to the solving the structure of Fusa, crystallisation of Fusa in complex with Fer_{ara} and in complex with PM1_{fer} was attempted. Fusa at 15 mg.ml⁻¹ in a buffer containing 50 mM Tris, 200 mM NaCl and 1 % β-OG pH 7.9, was mixed with an equal volume of the ferredoxin at 5 mg.ml⁻¹ in 50 mM Tris, 200 mM NaCl pH 7.9. Despite screening approximately 800 conditions, no crystals were obtained.'

4. For most of the paper, it is well written and the data are clearly presented (some minor suggestions for improvement are listed below). However, the last part of the Results section of the manuscript that describes data in support of the claim of iron piracy through plant ferredoxin uptake by this TBDR, is not of the same high level as the previous experiments and not convincing as a validation of the proposed TBDR-mediated iron uptake. For a panel of *Pectobacterium* strains, it was found that growth was promoted, to variable extents (only roughly scored as weak/medium/strong, depending on the strain), in iron-limiting medium supplemented with plant ferredoxins (three different types, also differing in their growth enhancement capacities). This observation is not really novel since such strain-dependent growth enhancement was described previously by the authors, albeit with spinach ferredoxin (ref. 15).

Our intention with this section was not to provide evidence in support of the role of the *Fus* operon in iron-acquisition from ferredoxin (this is well supported by our previous results, see response to Point 6). The intention was to determine if there is a difference in the ability of ferredoxins from diverse plant hosts to support growth of *Pectobacterium* spp. under iron limiting conditions. While these data only allow general conclusions to be drawn we think it is of interest, as the extent of growth enhancement is likely due at least in part to variation in the *FusA*-Ferredoxin binding interface outlined in previous sections. Subsequent to this (Ref 15) we knew only that spinach ferredoxin, but not human ferredoxin (adrenodoxin) is able to support growth on iron limiting media and block pectocin activity through competition for the receptor. Although our current studies on specificity towards ferredoxins from different plant species are not intended to be comprehensive they do begin to show that there is a level of specificity, with ferredoxins from *Arabidopsis* and potato, which are in the known host range for *Pectobacterium* supporting growth for a number of strains under iron limiting conditions. Conversely, ferredoxin from maize, which is not a known host for *Pectobacterium* spp, does not generally support growth. We feel that this data is in fact novel and expands significantly upon that presented in ref. 15 and sets the stage for future experiments that will aim to thoroughly dissect the molecular mechanisms of *FusA*-Ferredoxin binding. Interestingly in our initial follow up studies we can show that the ability of these plant ferredoxins to protect against pectocin M1 activity in competition assays is also variable with *Arabidopsis* ferredoxin offering more protection than potato ferredoxin and maize ferredoxin offering little protection. This supports the idea that altered *FusA*-ferredoxin interactions are responsible for the different levels of growth enhancement observed with different plant ferredoxins (see figure below).

In this experiment, plant ferredoxins at the concentrations indicated and pectocin M1 (18 μM) are spotted onto a growing lawn of *Pba* LMG2386. The clear zones indicate pectocin M1 mediated killing. The ferredoxins are spotted to the left of the pectocin and asymmetry in the zone of killing shows that pectocin killing is inhibited due to competition for the receptor.

5. The Methods section refers to a concentration range of iron chelator that was used to vary iron availability (L356) but it is not clear to which condition the data in Table S4 actually correspond.

Not including the concentration of iron chelator (200 μM) used was an oversight and this has now been included the figure legend for Table S4.

6. Although this experiment indicates that plant ferredoxin-derived iron becomes available to (some of) the pectobacteria, it does not unequivocally prove that TBDR-ferredoxin-mediated iron piracy is involved. For instance, can it be excluded that the ferredoxins are first degraded and the released iron would be taken up by the conventional (siderophore-dependent) route?

We have previously shown that plant ferredoxin and pectocins utilize the same receptor through competition assays (Ref 15). In these assays the addition of plant ferredoxin, but not

human ferredoxin, blocks pectocin mediated killing. This is only possible through competition for a common receptor. As would be expected if only plant ferredoxin were able to bind to the receptor, it is only plant and not human ferredoxin that is able to support growth on iron limited media. Additionally, we also showed in this publication that pectocin M1 is able to block growth enhancement by spinach ferredoxin in a *Pectobacterium* strain that is resistant to pectocin M1, showing that the receptor (which we now know is FusA) plays a key role in iron acquisition from ferredoxin. **These previously published data fully support the assignation of FusA as the ferredoxin receptor.** It is worth saying that these types of competition assays have been used for decades in the colicin field and are a well understood and validated method of showing competition for a common receptor.

6. As described earlier for spinach ferredoxin (ref. 15), the observed outcome ranges from lack of growth to well supported growth of individual strains, which may be expected due to the sequence divergence between plant ferredoxins. It is not clear what conclusive information regarding the proposed iron piracy strategy can be gained by such growth enhancement assay with multiple strains. For (most of) these strains, the TBDR sequence of interest is not known and, hence, sequence variation at the TBDR interface cannot be mapped on them to find possible correlations with corresponding sequence variation localized in the ferredoxins. As such, Figure 6 does not add much to the manuscript by mapping sequence divergence in three plant ferredoxins and the ferredoxin domain of pectocin M1 (there is some redundancy with ferredoxin comparison in Fig. 5D).

We accept the reviewer's view that our mapping of sequence variation between ferredoxins into the ferredoxin surface is premature without specific information on which residues are important for FusA binding and more information about sequence differences between FusA variants. As such, and in line with comments from reviewer 3, we have removed this portion of Figure 6 and 5D from the manuscript.

7. If the TBDR specifically supports the acquisition from plant ferredoxin-carried iron, one expects that this acquisition would be abolished in a TBDR-lacking mutant (concomitantly to becoming immune to the pectocin). Such immunity was demonstrated in this work, using the TBDR-mutant constructed for strain LMG 2386, but surprisingly, this mutant was not included in the growth enhancement experiment to compare its phenotype with the parent strain.

The observed growth enhancement of LMG2386 by plant ferredoxins is very weak and as such we cannot reliably use this strain and its FusA mutant pair to assess differences in growth enhancement by ferredoxin. We feel the using pectocin M1 sensitivity is a reasonable proxy given that it possesses a ferredoxin domain and we can show direct competition between Arabidopsis ferredoxin and pectocin M1 in our agar overlay assay. We attempted to generate FusA mutants in other *Pectobacterium* strains, but for reasons unknown and despite much effort we were not successful and as such elected to proceed with the LMG2386 mutant only.

8. Have the authors also tried to compare growth enhancement of strain LMG 2386 and its TBDR-deficient mutant by supplementation with the recombinant ferredoxin domain of pectocin M1 (should be a good interaction partner and could be a potential iron source)? Notably, growth enhancement of this strain (initially used to fish out the pectocin M1 target) and also of the strain for TBDR structure elucidation (SCRI1043) is only weak and evident for only one/two of the three ferredoxins tested. Could it be that the identified TBDR prefers other (plant) ferredoxins?

We agree with the reviewers comment and feel it is likely that LMG 2386 and SCRI1043 prefer ferredoxins from different plant species (or different ferredoxin isoforms). It will be interesting in future work to determine which ferredoxins these strains have a preference for. The effect of the pectocin M1 ferredoxin domain growth of *Pectobacterium* strains is cryptic. It doesn't appear to enhance growth and in some strains the individual ferredoxin domain displays limited cytotoxicity. When pectocin M1 is purified, while still red/brown in colour, the intensity of its colour is less than

pectocin M2 at the same concentration. Additionally, pectocin M1 ferredoxin domain runs lower on SDS page gel than plant ferredoxin or the pectocin M2 ferredoxin domain of the same size. It is tempting to speculate that as the pectocin M1 ferredoxin domain no longer needs to be a functioning ferredoxin, it is in the process of evolving cytotoxicity in its own right and losing its iron sulphur cluster in the process. These speculations will be investigated in future work.

9. Demonstration of abolished or strongly diminished acquisition of iron (isotope form) from iron-loaded ferredoxin, externally supplied to the TBDR mutant cells, would be more convincing than the observed growth enhancements.

We appreciate the reviewer's suggestion and agree this experiment would be a convincing way of demonstrating iron acquisition. However, we feel that performing this experiment is non-trivial, particularly in the LMG2386 mutant strain, which shows poor enhancement of growth with ferredoxin, and is outside the scope of the current study.

10. Apparently, there is also strong sequence divergence between the ferredoxin domains of the pectocins M1 and M2. It would be informative to include the pectocin M2 domain in sequence comparisons. Although most experimental work refers to pectocin M1 and its domain, the available crystal structure of pectocin M2 is introduced in the Results section on docking (L176). It should be made clear to the reader that pectocin M2 is not just an orthologue of pectocin M1 but is quite different from pectocin M1. The M1 and M2 ferredoxin-like parts share only about 60% amino acid sequence identity, which is in the same range of sequence divergence as between the M1 domain and the three plant ferredoxin used. Such differences likely have an impact on the respective interaction interfaces. Have the authors attempted to isolate the receptor for pectocin M2 (with crystal structure known) using the same approach that was successful for pectocin M1?

While we have extensively contrasted the differences between pectocin M1 and M2 in previous work^{1,2}, we fully agree with the reviewer's insight here and have included the pectocin M2 ferredoxin sequence for comparison in Figure 6. This is an interesting point as it's not clear that pectocin M1 and M2 were created by the same recombination event. Their genetic context is quite different and there is also considerable divergence in the linker region between the ferredoxin and cytotoxic domains. In a previous paper we also described Pectocin P (a fusion between a lysozyme like domain and a plant-like ferredoxin domain)², so it's well within the realm possibility that pectocin M1 and M2 have distinct ancestry. Regrettably, we only managed to obtain crystals and solve the structure of pectocin M2 so we are unable to contrast differences between the two proteins on a structural level. As the reviewer is no doubt aware one must often take what they can get from X-ray crystallography. We focused on pectocin M1 rather than pectocin M2 for this study, because pectocin M2 is a much less cytotoxic than pectocin M1, against our strains. Pectocin M2 originates from a *Pectobacterium brasiliensis* strain, which may explain its low level of activity against our *P. carotovorum* and *P. atrosepticum* isolates. Despite this, pectocin M2 still uses FusA as its receptor (presumably a variant different from FusA in the strains we used in the study, is required for full activity), in Figure S1 we show that pectocin M2 is capable of pulling down FusA when recombinantly expressed in *E. coli* membranes.

11. In the Discussion, the authors claim that by the discovery of FusA they have demonstrated for the first time that phytopathogenic bacteria are capable of obtaining iron from host proteins (L210-213; L249-251). Although such iron piracy is likely to occur, based on the results presented here, it has not been unequivocally demonstrated. Therefore these statements should be toned down and the title should be modified to reflect this.

With hindsight, these statements were poorly worded. For the first statement, the novelty is not that phytopathogenic bacteria are capable of obtaining iron from host proteins (we had already shown this) but that they do this through a TonB-dependent receptor. We have changed this statement to: 'The discovery of FusA demonstrates for the first time that, like their mammalian pathogenic counterparts, phytopathogenic bacteria can use a TBDR to specifically target iron containing host

proteins'. The second statement has been altered to: 'In summary this study represents first report and structural characterisation of a TBDR receptor that binds to iron-sulphur cluster containing ferredoxin domains.'

12. The distinction between the (plant) ferredoxin protomers and the ferredoxin domain that is part of the pectocin M proteins (fused to an extra killing domain) is often not clear from the description and nomenclature used. The latter is also designated as a 'ferredoxin' whereas it is actually a ferredoxin-like domain (even when recombinantly expressed as a separate polypeptide it should still be clear to the reader that this is derived from a bi-domain protein). This ambiguity in the descriptions is maintained in the titles: L87, L117, 'ferredoxin receptor' refers actually to the pectocin M1 receptor (this may give the false impression that already at that stage it has been proven that the receptor functions in binding/uptake of plant ferredoxins); L148, 'globular ferredoxin' is used for both plant protein and recombinant bacteriocin domain. It may be helpful for the reader to distinguish both by adopting different abbreviations for the ferredoxin domains and the 'genuine' ferredoxins. For instance: keep PM1(fer), with the subscript specifying the domain; instead of the Plant(fer) designation Fer(plant) may be used, with the subscript referring to the species origin (preferably derived from the Latin species name). Also note that apparently two different designations are used for the domain of pectocin M1 (L483, L486, Figure 4 also refer to it as MPC1(fer), but this is nowhere defined in the text).

We acknowledge that it would be useful to clearly define the difference between plant ferredoxin and the pectocin M1 ferredoxin domains and have modified the manuscript according to the reviewers suggestion i.e. We have replaced the Plant(fer) designation with Fer(plant). We feel that it would be simpler for the reader if we keep the plant designation as the common name as we use common names for potato and maize throughout the manuscript (we have included the latin name when we first mention these ferredoxins in the manuscript. We have highlighted these changes in the text. As stated above previous competition assays show that the spinach ferredoxin and pectocin M1 use the same receptor. To explain our logic in designating FusA the ferredoxin receptor, we have replaced the statement 'Thus, FusA is the ferredoxin receptor' with 'Thus, FusA is the receptor for the ferredoxin domain containing bacteriocin pectocin M1. As we have previously shown that pectocin M1 and spinach ferredoxin compete for binding to the same receptor, FusA is also a plant ferredoxin receptor ¹⁵. Prior to this we have changed 'ferredoxin receptor' to 'pectocin M1 receptor' in the initial stages of the manuscript.

13. The bioinformatic analysis shows that FusA represents a novel type of TBDR that is also found in several mammalian pathogens. Several additional gamma-proteobacteria are mentioned in the text but it is not highlighted that the analysis also identifies some important pathogenic beta-proteobacteria (like Neisseria, Bordetella) and epsilon/delta-proteobacteria (Campylobacter). The FusA-related proteins are collectively described as homologues but it should be pointed out that the sequence identities of many of these proteins is low (for most representatives listed in Table S1 sequence identity is less than 25%). As such, the nature of the substrates may also be quite different. The conclusion drawn from this analysis (L113-115 of Results) is rather speculative and more appropriate for the Discussion section.

We agree with the reviews comments and have modified the text to references the beta/epsilon/delta-proteobacteria our analysis identified (lines 112-114). We are unsure about the origin of the referees comment about the identification of FusA in Neisseria sp., as Neisserial homologues of FusA did not appear in our analysis. We have also modified the text to reflect the more distant evolutionary relationship between FusA-related proteins (lines 110-111. Despite low sequence identity, based on predicted secondary structure and the genetic linkage to *fusC* it is highly likely that the proteins identified in Table S1 share a common ancestry and so are homologues, in the broad sense of the term which was the meaning we intended.

Minor remarks

L66-69. Strictly speaking, bacteriocins like colicins, pectocins, ... are not only used for intraspecies competition (acting against strains of the same species), since examples are known (including pectocins studied here) where producer and target belong to different species (but usually belonging to the same genus).

We have changed this sentence to 'Colicin-like bacteriocins are protein toxins produced by Gram-negative bacteria mostly for intraspecies or intragenus competition and often parasitise TBDRs to gain entry to their target cell.'

When ferredoxin sequences are used for a particular plant species the corresponding database accession number /unique identifier should be provided (presence of isoforms).

We acknowledge this point and have modified the text accordingly (changes highlighted)

Figure S1. Mark lanes with numbers. Indicate the position of pectocin M1. Use same sequence identifier as in the text (PCC21_007820); Pba has not been defined. Correct: cartovororum.

We acknowledge this point and have modified the figure accordingly.

Figure S2. Represents data from a smaller (A) and a larger (B) subset and data in panels A and B thereby overlap. Cluster numbers used in panel A are not indicated in panel B. Why not use only the larger subset to avoid redundancy (B is said to "reinforce" A) and improve clarity of this figure?

The dataset utilised for clustering analysis in Figure S2, corresponds to the proteins listed in Table S1. We feel that this gives the reader the ability to visually assess the relationship between these proteins. These proteins were identified from the relatively well curated Uniprot rp75 sequence database. The sequences represented in B were identified from the NCBI RefSeq database; a large database with numerous redundant sequences. The aim with this panel was to confirm that the clustering identified in (A) is consistent with a larger dataset. Sequences from (A) are difficult to track in such a large dataset so they are not labelled. We feel that both panels are valuable to the manuscript.

Figure S6. The codes (FusA_1043 etc) used should be clarified in the text by indication of the source (species names/full strain numbers) of these FusA sequences in the legend

We acknowledge this point and have modified the figure legend accordingly (changes highlighted)

Figure S8. Enlarge panels A and B for better visibility. Panel C: is based on comparison of homologues in Figure S2, but the actual sequences used are not specified (those of small or large subset?, only those of a particular cluster including Pectobacterium?)

The figure legend for Figure S8 panel C should refer to the sequence alignment in Figure S5 (rather than Figure S2) this has been modified. The figure has been modified to enlarge panels A and B.

Table S1. No representative of Neisseria included?

No homologues of FusA from Neisseria were identified in our search of the Uniprot rp75 database. Our discussion of TBDR from Neisseria refers to the transferrin receptor TbpA. This protein isn't closely related to FusA, it is however an analogous example of iron piracy from a host protein.

Table S4. Rephrase sentence of legend: "... a qualitative measures the strength of the zone ..." (meaning not clear). Are data about the susceptibility of these strains to pectocin M1 and pectocin M2 available?

The term 'qualitative' was intended to acknowledge that these data don't provide a quantitative estimate of the growth enhancement effect. However, we have reworded it to remove ambiguity. Data about the susceptibility of these strains to Pectocin M1/M2 is provided in our previous work².

The effect is shown only for one strain at a single (high) concentration of the three plant ferredoxins (as an example in Figure 6.A). A better view of the concentration-dependent capacity to enhance growth would be provided by showing the extent of growth obtained with the different concentrations tested (dilution series). This could be presented as a multi-panel figure illustrating concentration-dependent growth profiles for these strains (or for representative strains with a particular profile).

We agree with the reviewers comment and have included a panel showing serial dilutions of the growth enhancement of due to ferredoxin for two strains in Figure 6.

Was the FusA protein expressed without the first 20 amino acids actually used in some of the experiments?

As outlined in the methods: FusA utilised for structural and NMR experiments was refolded from inclusion bodies produced in the cytoplasm of *E. coli*. This protein was expressed without the first 20 amino acids. This was because the first 20 amino acids represent a signal peptide which would direct the protein to the outer-membrane (and would be subsequently cleaved). The full-length protein was used for pull down experiment shown in figure S1.

The correct designations for proteins (FusA, ...; non-italic) and genes (*fusA*, ...; italic) should be used throughout the text and in figures.

We acknowledge this point and have modified the figures/figure legends accordingly.

It is important to always specify the strain that is used (not only in Methods but also in Results). For instance: the *P. atrosepticum* strain used for *fusA*-mutant construction (L96-98) is not mentioned; the strain from which the *fusA* gene was used for structural characterization is described without species designation as 'Pectobacterium SCRI1043' (L119; L383-384). Abbreviations used for bacterial (sub)species are not really standard (for instance 'Pba', 'Pbc'). They should be defined when the full name is used for the first time.

We have resolved these issues.

Reviewer #2

1. Given the fast exchange NMR data exhibited for PM1fer, is not possible to obtain an estimated KD of interaction between FusA and this ligand via a saturation experiment? Such information on the affinity of the complex would be useful. Also, the fact that 15N-labelled ARAfer exhibits slow exchange while 15N-labelled PM1fer exhibits fast exchange suggests a significant difference in binding affinity. Even if saturation experiments are not possible, perhaps the authors should comment further on this apparent affinity difference in the manuscript. Why does PM1fer bind with apparent lower affinity?

We would very much like to have demonstrated that the chemical shift perturbations seen for 15N labelled PM1fer titrated with FusA followed saturation kinetics (kinetics is the wrong word, but I'm stuck for the right one!). However, for this titration, this would have meant achieving high concentrations (20 mg/ml plus) of FusA. This presents two problems, first, such high concentrations of detergent-solubilised FusA are difficult to achieve and second, as saturation is approached, the PM1fer molecules would spend a higher proportion of the time bound to, and

tumbling as a FusA-sized particle. Under these conditions, it's very likely that the signals would relax too quickly to be observed.

We agree with the reviewer that the difference in apparent affinity of FusA for PM1fer and Arabidopsis fer is intriguing. There are multiple possible explanations for this observation including: Only the ferredoxin domain of pectocin M1 was used for this experiment and the linker region of the protein may be responsible for additional interactions that increase affinity. The FusA we were able to expression originated from SCRI1043 (we experienced difficulty with the expression of receptors from other strains), this strain is only weakly inhibited by pectocin M1, which could be explained by low affinity of pectocin M1 for FusA from this strain. Also, as we show that the outer-loops of FusA may close around the ferredoxin substrate, perhaps the when we perform these experiments in vitro (and this decouple from the protein motive force) we are only measuring the affinity of the FusA-Ferredoxin encounter complex, which may be inherently of lower affinity for the pectocins compared to plant ferredoxins. While we acknowledge that these data leave unanswered questions about the details of FusA/Ferredoxin interactions we plan to improve our understanding of this system in future work.

2. Lines 229-230: The authors claim that the Fus proteins "represent the first members of a family that plays an important role in the acquisition of iron from small iron-containing host proteins". This statement should be slightly re-written for clarity, perhaps changing 'small iron-containing host proteins' to 'ferredoxin-containing proteins'. There are other TBDRs in nature that bind to iron-loaded protein ligands at their extracellular loops. The TBDR HgbA from Haemophilus and Actinobacillus, for example, is a hemoglobin-binding protein that is predicted to have extensive extracellular loop regions forming a protein interaction interface to bind hemoglobin for iron extraction. In fact, the authors need to discuss FusA in the context of HgbA, since to my knowledge this is the only other TBDR known to bind to another protein as a ligand.

When chose to refer to 'small iron-containing host proteins' rather than 'ferredoxin-containing proteins', because the FusA homologues may not target ferredoxin as their ligand. Due to the low sequence identity between FusA homologues it is conceivable that they may have evolved to target something other than ferredoxin. We hypothesize that target for these homologues would still be small and iron containing, due to the association with homologues of the predicted periplasmic protease (Homologous to FusC) and the regulation of these proteins in response to iron limitation in a number of other studies mentioned in the text. A number of other TBDRs have been identified which bind an iron containing protein as a ligand, including the Transferrin and Lactoferrin receptors possessed by a number of species (Including Neisserial sp.) and hemophore receptors which are expressed by the bacteria in conjunction with secreted 'hemophore protein' which binds environmental heme and then docks with the receptor delivers it to the receptor. Like HgbA, these TDBR also interact with their ligand protein via extensive extra cellular loops. **The distinction we are attempting to make is that these receptors bind a ligand that is a rather large protein, of a size incapable of passing through the lumen of a TBDR.** In this work we elected to limit our discussion of protein binding TBDR to those for which structural characterization has been performed. However, we agree with the reviewer that HgbA represents an interesting example of a protein binding TBDR.

3. The authors describe FusA as a TonB-dependent receptor, yet go on to state that it interacts with the 'TonB-like' protein FusB. Just how similar is FusB in comparison with TonB? For example, from what I can see of the primary sequence (Fig. S6), FusA has a Ton Box sequence (53DTILV57, not resolved in the electron density). Is this also a FusB-binding sequence? The authors should elaborate on this. Does Pectobacterium also express TonB in addition to FusB?

The sequence alignment between FusB and TonB from *E. coli* is as follows:

membrane spanning protein in TonB-ExbB-ExbD transport complex protein [Escherichia cc
 Sequence ID: gb|AKK12988.1 Length: 244 Number of Matches: 1
▶ See 1 more title(s)

Range 1: 26 to 244 GenPept Graphics ▼ Next Match ▲ Previous Match

Score	Expect	Method	Identities	Positives	Gaps
99.0 bits(245)	9e-25	Composition-based stats.	53/240(22%)	88/240(36%)	21/240(8%)
Query 43	GSMQVTMMAAAMSQAADTSPPVADPPVAQPTPVVMPILTPLEHPNPVIKQPVIERKPV				102
Sbjct 26	GAVVAGLLYTSVHQVIELPAPAQPISVTMVTMPADLEPPQAVQPPPEPVVEPEPEPEPIPE				85
Query 103	PVTEKKPPVREKRPLEKKPEDKPEQQQARSQQTQQTQQAEEKKSESPVATAQTGDAPSPM				162
Sbjct 86	PPKEAPVVIE-----KPKPKPKPKPKPVKKVQEQQPKRDVKPVES				124
Query 163	PSSVGMPPGSASTAKAGESDSGQAVRGAGKSNQNFKALHRRVNYPSRAKALGVEGNVRVK				222
Sbjct 125	RPASPFENTAPARLTSSATAAATSKPVTSVASGPRALSRNQPYPARAQLRIEGQVKVK				184
Query 223	FDITGSGVTNVNQILSETPDGVFGDDVMKDMARWRYRTEAPVENQVVSIVFKLNGHIQVD				282
Sbjct 185	FDVTPDGRVDNVQILSAKPANMFEREVKNAMRRWRYEPGKPGSGIVVNILFKINGTTEIQ				244

The proteins have significant conservation, especially in the binding domain at the C-terminus. Pectobacterium has a number of other TonB homologues, however we are not aware of any studies characterizing their expression. It is common for some bacteria to possess multiple TonB homologues. Studies have been published showing these proteins interact with a specific subset of TBDRs in Pseudomonas and Serratia^{3,4}.

Our basis for describing FusA as a TBDR is structural homology to other well characterised TBDRs. We agree that 53DTILV57 looks like a Ton Box, but have not yet investigated this experimentally at this point. We chose not to discuss the role of FusB in this work as we are yet to perform any experimental work on it. This is a plan for future experiments however.

4. Figs S4, C&D: The electron densities shown in these figures appear quite smoothly distributed given the resolution. Are these 2Fo-Fc maps? The nature of the maps should be specified in the figure legend. Ideally, something like a simulated annealing omit map should be shown as a better assessment of quality of the diffraction data.

These images are from 2Fo-Fc maps, we agree that electron densities do appear quite reasonable for the resolution, which was pleasing and assisted in model building. We agree that a simulated annealing omit map is a better measure of the quality of the maps and so have replaced the 2Fo-Fc map with a SA-omit map and so have modified the figure accordingly.

5. Figure 5. This is a very busy figure with a lot going on in it. The stereodiagrams are too small to effectively portray anything of interest in 3D. The figure that could really benefit from a stereoview is Fig. 5A. In contrast, Fig 3B is required to show the overall barrel cross section in comparison with the diameter of the docked ferredoxin -- a 3D representation is not necessary for that. Furthermore, Fig. 5D uses the coloring scheme employed in Fig. 6 to indicate positions of sequence dissimilarity between related proteins -- it seems out of place in Fig. 5. I would suggest moving Fig. 5D into Fig. 6 (perhaps as Fig. 6D), and to convert Fig. 5B to a 2D figure. This would free up space to enlarge Fig. 5A

We acknowledge this point and agree with the reviewer. We modified figure 5 by removing panel D and the stereo view of panel B. We have also enlarged figure 5A to give a better view of this panel.

Minor points:

1. Line 249-250: "TBDR receptor" - the use of 'receptor' is redundant here.

We acknowledge this point and have modified the text accordingly

2. Fig 5.: the ferredoxin ligand is shown as a surface representation, which is effective; however, is this a van der Waals surface? It looks like one. The ligand should instead be rendered as a solvent-accessible surface in Fig. 5 to more accurately indicate the dimensions of the ligand relative to the barrel pore.

We respectfully disagree with the review on this point, based on the following rationale.

Solvent accessible surface is calculated (usually using a rolling ball type algorithm) to show the surface area that a H₂O molecule can sample/access on the surface of a protein. This is appropriate for showing the surface of a protein in solution, however what we are really interested in showing in this figure is the possibility of the ferredoxin fitting through the lumen of the FusA barrel. If the ferredoxin does indeed traverse the barrel, it will form interactions with the atoms on the inside of the barrel, these interactions will presumably not involve solvent. In my understanding a Van De Waals surface roughly represents the minimum distance that an atom can come to another atom. This surface seems more appropriate for what we are trying to show. My case study for this point is that if you display the plug domain of FusA using the solvent accessible surface it looks far too big for the barrel, with overlaps with the barrel when drawn as sticks. I feel that displaying the ferredoxin as solvent accessible surface gives the misleading impression that the ferredoxin is too large to fit through the barrel when in reality it's of the same dimensions as the plug domain, which we know from the crystal structure fits (snugly) inside the barrel.

As such we have elected to keep the Van Der Waals surface, but clearly state in the figure legend that this is the kind of surface we are using.

3. Fig. S5 C&D: Extracellular loops around M732 are clearly being stabilized by an NCS-related protomer in the asymmetric unit. What's stabilizing the other loops (i.e. the other half of the "catcher's mitt"? Crystal contacts?

There are additional crystal contacts around R381, T597 on MolA and R381, D611, Y658 on MolB. However, despite these interactions it would be our guess that these other side of the 'catcher's mitt' is much more stable, due to extensive inter-loop interactions and may not be greatly stabilised by these contacts.

Reviewer #3

1. Overall, well written and organized paper. A number of grammar errors but I am sure those will be fixed before being finalized.

2. The manuscript is short and concise, especially given there are many figures (including supp figs), but does a nice job of following up on the structural studies, etc.

3. In the pull down assay that discovered FusA, the bait (pectocin M1) is in far excess, yet 2 liters of a saturated culture was used to yield only a barely detectable band on an SDS-PAGE gel. The affinity for such interactions has historically be very high, so what does this say about the expression levels of FusA? Or the affinity? Did the authors ever develop an antibody against FusA to determine what percentage of the FusA they were actually capturing in this assay?

This experiment doesn't provide direct information about affinity (although later NMR experiments suggest a relatively weak affinity between the ferredoxin-like domain of PM1 and FusA). As the reviewer suggests, the lack of abundance of FusA in the initial pulldown experiment may reflect weak affinity for pectocin M1. Alternatively, as the reviewer suggests a weak band may be observed because very little receptor was present in the membranes. TBDRs are often expressed by bacteria very low levels (10s to 100s of copies per cell in some cases), even when induced. Furthermore, induction of FusA by growing Pectobacterium in iron limited LB is likely a poor mimic for the physiological environment in which this receptor would be expressed (presumably in

planta). Growth under these conditions may only induce FusA at a low level. We didn't have an antibody to FusA available during these studies.

4. The authors never show they can actually make a FusA:ferredoxin complex, nor do they indicate the possible binding affinities for one another? Therefore, for the NMR experiments, we are left to assume this, but I think some information here is needed since it hasn't technically be established in vitro using the refolded sample.

NMR is a highly sensitive and specific tool for detecting protein-protein interactions. As outlined below the NMR experiments in which purified FusA was titrated into ^{15}N labeled PM1_{fer} and Fer_{Ara} unambiguously show interaction between the ferredoxins and FusA. These data provide an indication of the affinity of FusA for these substrates based on the exchange rate of the complex. The fast exchange of FusA/ PM1_{fer} suggested weak interaction, generally complexes which exhibit fast exchange on the time scale of an NMR experiment have high μM affinity. For the FusA/ Fer_{Ara} complex, the slow exchange observed is characteristic of a tighter affinity complex, generally low μM or stronger. We feel that these data in conjunction with the other results presented in the paper are compelling evidence that a FusA:ferredoxin complex is formed.

Also, admittedly not an NMR expert, it seems odd to me that with increasing FusA, the peaks disappear almost entirely? What does this say about the fold of ferredoxin in the presence of FusA? What is happening to the structure of ferredoxin here to make the peaks disappear? If I had looked at the 1:1 complex spectra first, I would have concluded a lack of structure in ferredoxin when bound to FusA or that FusA is somehow unfolding ferredoxin. Can the authors clarify this?

The appearance of protein NMR signals varies with foldedness of the protein, rate of change between different conformations and critically, overall tumbling rate of the molecule in solution. In these experiments FusA is a 97 kDa protein solubilised in a detergent micelle producing a species of total MW around 117 kDa. Species of this size tumble slowly in solution meaning that their NMR spectra are effectively invisible without isotopic dilution of the ^1H nuclei with ^2H and the application of appropriate pulse sequences at (ultra)high magnetic field. The most likely explanation for the disappearance of the peaks originating from well ordered parts of the Fer_{ara} spectra is that as a larger proportion of Fer_{ara} becomes bound to FusA, it adopts the tumbling regime of the complex and its signals become undetectable due to fast relaxation. In contrast, signals from the disordered parts of the protein (purification tag, selected sidechains) remain visible since their motion is sufficiently decoupled from the rest of the protein that their local motion does not cause rapid relaxation. PM_{fer} -FusA complexes on the other hand will have only a short lifetime such that PM_{fer} will predominantly tumble with its uncomplexed correlation time.

5. I strongly suggest that the authors make the models of their protein complexes available as supplementary download so that others in the field can properly assess and utilize them as needed.

We agree with the reviewer on this point and have made the docked complex coordinates available as supplemental files

6. The authors did not follow up their modeling studies of the complex with some mutagenesis to determine if they can disrupt the binding or not. This would make the manuscript much better and more conclusive.

We are beginning to perform these studies and to determine the basis of the difference in affinities for different plant ferredoxins, but to do this properly is a paper in itself and so we think this is beyond the scope of the current manuscript.

7. The crystallographic table is missing some info; 'Mean/Wilson plot B-value'

Leaving out the Mean/Wilson plot B-values was an oversight, they have now been included in the table

a. Not necessary to put down percentiles for MOLPROBIT scores, those are relative and will not remain constant.

We have deleted these scores from the table

b. Ramachandran scores do not add up to 100% for potato ferredoxin

We have fixed the Ramachandran values for potato ferredoxin

c. In footnote c, what does 'sup 16' indicate?

This referred to the reference for MOLPROBITY, but was not updated from a previous version, as we have removed the MOLPROBITY values from the table this has been removed from the revised manuscript.

8. For the movie, it would be nice to include a title slide. Also, I suggest looping the morph such that it not only closes, but opens and maybe cycles a few times, and maybe even show this from a few different angles.

We agree with the reviewer's suggestion and have improved the movie to include a title page and initial view of FusA followed by the animation showing the opening and closing of FusA in the NMA simulations from different angles.

- 1 Grinter, R., Milner, J. & Walker, D. Ferredoxin containing bacteriocins suggest a novel mechanism of iron uptake in *Pectobacterium* spp. *PLoS ONE* **7**, e33033, doi:10.1371/journal.pone.0033033 (2012).
- 2 Grinter, R. *et al.* Structure of the atypical bacteriocin pectocin M2 implies a novel mechanism of protein uptake. *Molecular Microbiology* **93**, 234-246, doi:10.1111/mmi.12655 (2014).
- 3 Paquelin, A., Ghigo, J. M., Bertin, S. & Wandersman, C. Characterization of HasB, a *Serratia marcescens* TonB-like protein specifically involved in the haemophore-dependent haem acquisition system. *Molecular Microbiology* **42**, 995-1005, doi:10.1046/j.1365-2958.2001.02628.x (2001).
- 4 Zhao, Q. & Poole, K. A second tonB gene in *Pseudomonas aeruginosa* is linked to the *exbB* and *exbD* genes. *FEMS Microbiology Letters* **184**, 127-132, doi:10.1111/j.1574-6968.2000.tb09002.x (2000).

Reviewer #1 (Remarks to the Author):

Most of the comments made by Reviewer #1 have been appropriately addressed, but some issues remain.

1. The (unchanged) title still suggests that direct evidence (structural basis) has been obtained for iron piracy. However, this study primarily provides novel insight in the binding of an iron-containing host protein to a TBDR, corresponding to the initial stage of this phenomenon (uptake). The claim of iron piracy (actual use of the captured iron) is based on growth enhancement experiments (not on acquisition of isotopic iron, for instance)(also see replies 1.4, 1.7-1.9, 1.11). Such growth enhancement was reported previously for a particular plant ferredoxin and is here taken further by using three other ferredoxins derived from host and non-host plants. The revised figure 6 suggests that host specificity may be involved in plant ferredoxin-dependent growth stimulation. Unfortunately, the strain used for receptor identification and construction of a TBDR-receptor mutant shows only poor growth enhancement and it said not to be suitable for comparing wild type and mutant in the qualitative growth assay on solid medium; maybe quantitative differences could be demonstrated for growth curves in liquid medium?. Better (but still indirect) support for the iron piracy would be provided if a strongly growth-promoted strain could be shown to be affected in its capacity to use the ferredoxin iron when it lacks its TBDR (authors could not yet obtain such crucial mutant; see reply 1.7).

In my opinion, a modified (sub)title better reflecting the results of the study would refer to the receptor interaction (and also specify that it involves plant ferredoxins).

2. Minor point. The added value of showing two CLANS plots in Figure S2 is not clear; somehow even confusing since annotation of the corresponding clusters is not always consistent (see reply 1.13 and related minor comment). Some clusters are present/labeled in only one of both panels (for instance, *Achromobacter* absent from panel B; *Acinetobacter* not present in panel A). The authors did not yet include some related bacterial TBDRs, e.g. from *Neisseria* (see for instance AKG09_09800 and homologues with >35% amino acid identity to *FusA*; not included in Table S1).

Reviewer #2 (Remarks to the Author):

The authors have adequately addressed comments raised, and have revised their manuscript appropriately. I recommend no further revisions.

Dear Rebecca

Thank you again for the comments on our paper now titled 'Structure of the bacterial plant-ferredoxin receptor FusA', NCOMMS-16-01698-T. We have addressed the reviewers' insightful comments and this has again much improved the presentation of the manuscript. Please find our responses to the specific comments of the reviewers below. We have also highlighted the major changes described in the manuscript. We hope these changes will make our paper suitable for publication in Nature Communications.

Yours sincerely

Dan Walker and Rhys Grinter

Reviewers' comments:

Reviewer #1 (Remarks to the Author):

Most of the comments made by Reviewer #1 have been appropriately addressed, but some issues remain.

1. The (unchanged) title still suggests that direct evidence (structural basis) has been obtained for iron piracy. However, this study primarily provides novel insight in the binding of an iron-containing host protein to a TBDR, corresponding to the initial stage of this phenomenon (uptake). The claim of iron piracy (actual use of the captured iron) is based on growth enhancement experiments (not on acquisition of isotopic iron, for instance)(also see replies 1.4, 1.7-1.9, 1.11). Such growth enhancement was reported previously for a particular plant ferredoxin and is here taken further by using three other ferredoxins derived from host and non-host plants. The revised figure 6 suggests that host specificity may be involved in plant ferredoxin-dependent growth stimulation. Unfortunately, the strain used for receptor identification and construction of a TBDR-receptor mutant shows only poor growth enhancement and it said not to be suitable for comparing wild type and mutant in the qualitative growth assay on solid medium; maybe quantitative differences could be demonstrated for growth curves in liquid medium?. Better (but still indirect) support for the iron piracy would be provided if a strongly growth-promoted strain could be shown to be affected in its capacity to use the ferredoxin iron when it lacks its TBDR (authors could not yet obtain such crucial mutant; see reply 1.7).

In my opinion, a modified (sub)title better reflecting the results of the study would refer to the receptor interaction (and also specify that it involves plant ferredoxins).

We appreciate the reviewers desire to see the growth enhancement due to ferredoxin directly linked to the presence/absence of FusA. We attempted many experiments to show this using LMG2386, including liquid culture growth curve experiments. However, we don't consider the results definitive enough for publication. We also attempted to make the 'critical knockout' suggested by the reviewer, however this proved more difficult than

anticipated. We will pursue this goal in future work however. We concede that the claim of iron piracy in our original title might be overstating the data presented in this work, as such we have modified the title to reflect more directly the data presented in the paper. The modified title is '**Structure of the bacterial plant-ferredoxin receptor FusA**'.

2. Minor point. The added value of showing two CLANS plots in Figure S2 is not clear; somehow even confusing since annotation of the corresponding clusters is not always consistent (see reply 1.13 and related minor comment). Some clusters are present/labeled in only one of both panels (for instance, *Achromobacter* absent from panel B; *Acinetobacter* not present in panel A). The authors did not yet include some related bacterial TBDRs, e.g. from *Neisseria* (see for instance AKG09_09800 and homologues with >35% amino acid identity to FusA; not included in Table S1).

In reference to this point we have repeated the bioinformatics analysis behind this figure and recompiled Table S1 and redrawn Figure S2. When we initially performed the HMMER search for Table S1 and Figure S2 the RP75 database didn't contain the *Neisseria* sequences referenced by the reviewer. However, they have since been included in this database and so are now present in our analysis (along with a number of new sequences). We feel that this new analysis is more representative of the distribution of FusA homologues (114 sequences compared to 43). As such and in order to avoid the confusion pointed out by reviewer 1 we have removed panel B from figure S2. We feel that these changes have improved the manuscript and appreciate the reviewers continued attention to this point.

Reviewer #1

(Remarks to the Author)

The authors have changed the title and this better reflects the results of the study.

The analysis of the distribution of homologues has been updated and this provides an improved overview of the broader occurrence (and possible significance) of this system.